# *Salmonella* actively modulates TFEB in murine macrophages in a growth-phase and time-dependent manner

Subothan Inpanathan,[1,2] Erika Ospina-Escobar,[1,2] Vanessa Cruz Li,[1] Zainab Adamji,[1,2] Tracy Lackraj,[1] Youn Hee Cho,[1,2] Natasha Porco,[1,2] Christopher H. Choy,[1,2] Joseph B. McPhee,[1,2] Roberto J. Botelho[1,2]

**ABSTRACT** The transcription factor TFEB drives the expression of lysosomal, autophagic, and immune-responsive genes in response to LPS and phagocytosis. Interestingly, compounds that promote TFEB activity enhance bactericidal activity, while intracellular pathogens like *Mycobacterium* and *Salmonella* repress TFEB. However, *Salmonella enterica* sv. Typhimurium (*S*. Typhimurium) was reported to actively stimulate TFEB, implying a benefit to *Salmonella*. To better understand the relationship between *S*. Typhimurium and TFEB, we assessed if *S*. Typhimurium regulated TFEB in macrophages in a manner dependent on infection conditions. We observed that macrophages that engulfed late-logarithmic grown *Salmonella* accumulated nuclear TFEB, comparable to macrophages that engulfed *Escherichia coli*. In contrast, stationary-phase *S*. Typhimurium infection of macrophages actively delayed TFEB nuclear mobilization. The delay in TFEB nuclear mobilization was not observed in macrophages that engulfed heat-killed stationary-phase *Salmonella*, or *Salmonella* lacking functional SPI-1 and SPI-2 type 3 secretion systems. *S*. Typhimurium mutated in the master virulence regulator *phoP* or the secreted effector genes *sifA*, and *sopD* also showed TFEB nuclear translocation. Interestingly, while *E. coli* survived better in *tfeb*[−/−] macrophages, *S*. Typhimurium growth was similar in wild-type and *tfeb*[−/−] macrophages. Moreover, *Salmonella* survival was not readily affected by its growth phase in wild-type or knockout macrophages, though in HeLa cells late-log *Salmonella* benefitted from the loss of TFEB. Priming macrophages with phagocytosis enhanced the killing of *Salmonella* in wild-type, but not in *tfeb*[−/−] macrophages. Collectively, *S*. Typhimurium orchestrate TFEB in a manner dependent on infection conditions, while disturbing this context-dependent control of TFEB may be detrimental to *Salmonella* survival.

**IMPORTANCE** Activation of the host transcription factor TFEB helps mammalian cells adapt to stresses such as starvation and infection by upregulating lysosome, autophagy, and immuno-protective gene expression. Thus, TFEB is generally thought to protect host cells. However, it may also be that pathogenic bacteria like *Salmonella* orchestrate TFEB in a spatio-temporal manner to harness its functions to grow intracellularly. Indeed, the relationship between *Salmonella* and TFEB is controversial since some studies showed that *Salmonella* actively promotes TFEB, while others have observed that *Salmonella* degrades TFEB and that compounds that promote TFEB restrict bacterial growth. Our work provides a path to resolve these apparent discordant observations since we showed that stationary-grown *Salmonella* actively delays TFEB after infection, while late-log *Salmonella* is permissive of TFEB activation. Nevertheless, the exact function of this manipulation remains unclear, but conditions that erase the conditional control of TFEB by *Salmonella* may be detrimental to the microbe.

**KEYWORDS** bacteria, macrophages, transcription factors, cell adaptation, innate immunity, *Salmonella*, culture, lysosomes

Address correspondence to Roberto J. Botelho, rbotelho@torontomu.ca.

Subothan Inpanathan and Erika Ospina-Escobar contributed equally to this article. Author order was determined based on seniority of degree, whereby S. Inpanathan is a current 5th year Ph.D. student and E. Ospina-Escobar was a 2nd year M.Sc. and is currently not in the scientific career path.

The authors declare no conflict of interest.

See the funding table on p. 25.

Macrophages are innate immune cells that hunt and engulf microbes through phagocytosis. As a corollary, following phagocytosis, microbes are quarantined within phagosomes, which are then transformed into hostile phagolysosomes. During this maturation process, phagosomes acquire a microbicidal milieu enriched in hydrolytic enzymes, antimicrobial peptides, and an acidic and oxidative lumen (1–4).

Phagosome maturation is critically dependent on the endolysosomal membrane system. Phagosomes fuse sequentially with endosomes and ultimately with lysosomes in a process that depends on various Rab GTPases, lipid mediators like phosphatidylinositol-3-phosphate, and motor and SNARE proteins (2–10). Interestingly, while phagosome-lysosome fusion has been known for decades (11, 12), it was generally assumed that lysosomes and their properties remained unchanged during phagocytosis, phagosome maturation, and interaction with microbes—though at least one study in the 1970s suggested that phagocytosis in macrophages induced lysosomal enzyme levels (13). However, we now understand that lysosomes are remodeled in response to various stresses, including interaction with microbes, microbe-associated molecular patterns (MAMPs), and phagocytosis (14–22). In part, this is achieved by activation of the transcription factor TFEB, which drives the expression of endolysosomal, autophagy, and immunomodulatory genes (17–20, 23, 24).

TFEB is activated by numerous signals including amino acid deprivation, metabolic stress, mitochondrial stress, lipid metabolism, and several immune stimuli such as phagocytosis of IgG-coated particles, several types of bacteria, and pattern recognition receptors that bind MAMPs (17, 18, 23–32). TFEB activation increases intracellular killing of microbes like *Escherichia coli*, *Mycobacterium tuberculosis*, *S.* Typhimurium, *Staphylococcus aureus*, and *C. albicans* (18, 33–35). Overall, TFEB activation is thought to favor immune function and repress microbe survival (19, 20, 36). However, there is evidence for some microbes actively manipulating TFEB function.

Macrophages that engulfed living *S.* Typhimurium displayed nuclear TFEB translocation relative to macrophages that internalized non-viable *S.* Typhimurium in a process dependent on PLC-PKD pathway (30). A more recent study by the same authors indicates that dead *Salmonella* can also activate TFEB, but more slowly (22). Nevertheless, the intentional stimulation of TFEB by invading microbes is peculiar since TFEB stimulation is typically thought to benefit the host. Indeed, additional studies suggest that ectopic activation of TFEB with compounds like VX-765, acacetin and 4-octyl itaconate can limit *Salmonella* survival (33, 34, 37). Collectively, these observations suggest a complex interplay between TFEB and *S.* Typhimurium that requires further investigation.

*S.* Typhimurium is a microbe that hijacks mammalian cells through a complex virulence program involving two type III secretion systems (T3SS), SPI-1 and SPI-2, that collectively inject ~50 bacterial effector proteins into infected host cells that manipulate the behavior of both epithelial and immune cells (38–40). The SPI-I T3SS is usually associated with invasion of epithelial cells, a bacteria-driven process that remodels the epithelial cell plasma membrane to ruffle and internalize the bacteria (40–45). *S.* Typhimurium can also infect macrophages during macrophage-driven phagocytosis (46, 47). Here, *S.* Typhimurium does not have to be in an invasive state *per se*, but the SPI-2 T3SS is essential to usurp, remodel and even escape the *Salmonella*-containing vacuole to promote its replication within the host macrophage (39, 46–48). Importantly, the behavior of *Salmonella* within a host cell may depend on both bacterial and host cell conditions including entry mode, growth and metabolic state, and time post-infection (41, 47, 49–52). In fact, the growth state and nutritional conditions of *Salmonella* appear to modulate the entry mode, where late-log/early stationary phase grown *Salmonella* favor SPI-1 expression, while late stationary growth primes SPI-2 expression (50, 53, 54).

Given this, we postulated that *Salmonella* and TFEB activation may depend on the invasive state of *Salmonella* and/or bacterial growth conditions. Here, we provide evidence that *Salmonella* grown to stationary-phase actively and transiently represses TFEB in macrophages. In comparison, *Salmonella* grown to late log-phase promotes early translocation of TFEB to the nucleus. Moreover, while TFEB is needed to suppress *E. coli*

survival in macrophages, its deletion had no apparent effect on *Salmonella* growth in macrophages. Yet, pre-activation of TFEB by phagocytosis increased the killing of both bacterial species. Overall, we reveal a complex interplay between TFEB and *Salmonella* infection conditions.

## RESULTS

### *Salmonella* grown to stationary phase delays activation of TFEB after uptake by macrophages

Phagocytosis of IgG-coated particles, unopsonized *E. coli*, *M. smegmatitis*, *and S. aureus*, and invasive *Salmonella* all activate TFEB in macrophages (17, 18, 22, 24, 30). In most cases, TFEB activation is thought to boost the immunological response by stimulating lysosomal, autophagy, and immune-protective genes (17–19, 24). However, in the case of invasive *Salmonella*, activation of TFEB has been reported to be driven by *Salmonella*, which implies a benefit to the microbe during the nvasion of macrophages (30). In apparent conflict with these observations, there are also observations showing that *Salmonella* represses TFEB and that TFEB stimulation increases *Salmonella* killing (33, 34). Given these discrepancies, we sought to better understand the relationship between *Salmonella* infection and TFEB activation.

We began by comparing the ability of *Salmonella* grown to stationary phase to elicit TFEB activation in macrophages by measuring the relative nuclear to cytosolic (N/C) ratio of TFEB. Stationary-phase *Salmonella* cultures are reported to be enriched in non-invasive *Salmonella*, and like *E. coli*, engulfment is thought to be driven by macrophage phagocytosis rather than by invasion (47, 55). As expected, we found that phagocytosis of non-pathogenic *E. coli* DH5α increased the N/C ratio of GFP-TFEB in RAW macrophages within 1 h of phagocytosis relative to resting macrophages (Fig. 1A and B). By contrast, the N/C ratio of GFP-TFEB in macrophages that engulfed stationary-phase *Salmonella* was similar to resting cells and significantly lower than macrophages that engulfed *E. coli* (Fig. 1A and B). Moreover, phagocytosis of heat-killed *Salmonella* previously grown to stationary-phase elicited nuclear mobilization of GFP-TFEB that was comparable to macrophages that engulfed *E. coli* (Fig. 1A and B). Collectively, these observations suggest that stationary-phase *S.* Typhimurium actively represses TFEB nuclear translocation in macrophages at 1 h post-infection. However, we also observed an increase over time in the N/C ratio of GFP-TFEB in macrophages that engulfed stationary-phase *Salmonella*; at 4 and 6 h post-engulfment, the N/C ratio of GFP-TFEB in these cells was higher than resting cells or those at 1 h post-infection, and comparable to macrophages infected with *E. coli* (Fig. 1C and D). Thus, stationary-phase *Salmonella* delays TFEB mobilization to the nucleus, rather than fully arresting it.

### *Salmonella* growth conditions affect TFEB response in macrophages

The growth phase of *Salmonella* is thought to affect the expression of invasion effectors and modulate the mode of infection (41, 47, 50). Typically, *Salmonella* is grown to late log-phase before experimentation. We thus tested whether the growth phase of *Salmonella* affected the N/C ratio of TFEB in RAW cells. First, we observed that stationary-phase and log-phase *E. coli* similarly induced nuclear translocation of TFEB (Fig. 2A and B). Moreover, TFEB remained enriched in its nuclear distribution in macrophages 1 or 4 h post-uptake of *E. coli* grown to either culture phase (Fig. 2A, C, and D). *Salmonella*-infected cells displayed a contrasting behavior in terms of TFEB nuclear entry. While stationary-phase *Salmonella* failed to activate TFEB at 1 h post-infection, as illustrated before, macrophages infected with late-logarithmic *Salmonella* significantly mobilized TFEB into the nucleus within 1h of uptake (Fig. 2A and B), consistent with observations by Najibi et al. (30). We observed similar results when staining for endogenously expressed TFEB; stationary-phase live *Salmonella* was the only condition that failed to enrich TFEB in the nucleus after 1 h of uptake, while live and dead *E. coli*, live late log-phase *Salmonella*, and non-viable stationary-phase and late log-phase *Salmonella* all triggered accumulation of TFEB in the nucleus relative to resting cells (Fig. 3A and B).

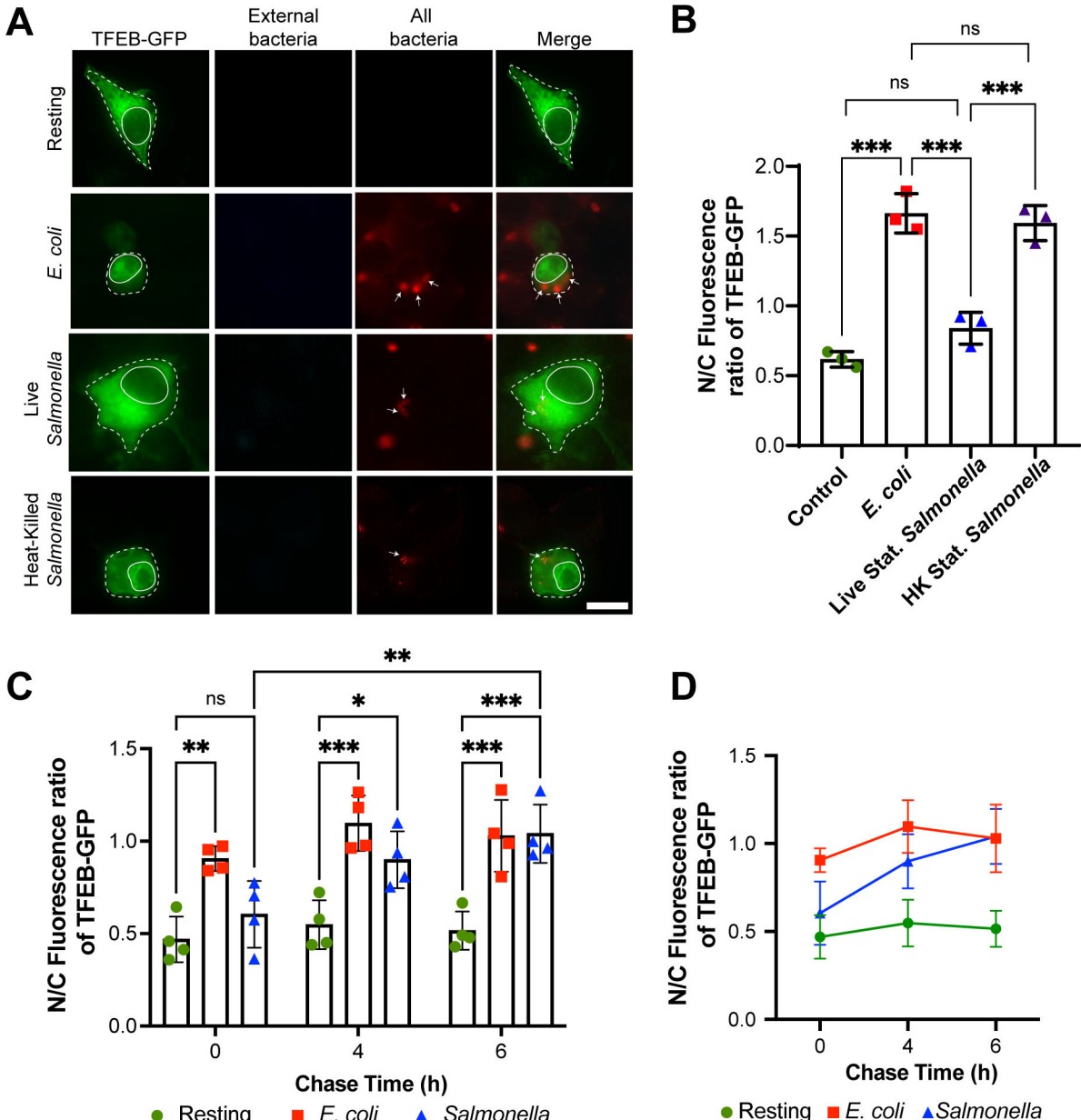

**FIG 1** Stationary-grown *Salmonella* delays TFEB nuclear translocation after phagocytosis. (A) Epifluorescence micrographs of fixed RAW macrophages transiently expressing GFP-TFEB and that were given no bacteria (resting) or incubated for 1 h with living *E. coli*, or with living or heat-killed *Salmonella* grown to stationary phase. Samples were stained for external (before permeabilization) and all bacteria (after permeabilization) using anti-*E. coli* or anti-*Salmonella* antibodies. Arrows indicate internalized bacteria. Scale bar = 10µm. (B) The nuclear to cytosolic (N/C) fluorescence ratio of GFP-TFEB in macrophages was quantified for the indicated conditions after 1 h of uptake. Means ± SEM was tested using one-way ANOVA test and post hoc Tukey's test, where *** indicates $P < 0.001$. (C and D) GFP-TFEB transfected RAW macrophages engulfed live stationary *E. coli* or *Salmonella* for 1, 4, or 6 h. The N/C fluorescence ratio of GFP-TFEB was then quantified. (C and D) are different representations of the same data. Data are shown as mean ± SEM from three independent (B) or four independent (C, D) experiments, scoring at least 50 cells for each condition per experiment. For (C and D), mean ± SEM are shown and tested using two-way ANOVA test and post hoc Tukey's test, where *, **, and *** indicate $P$ values of 0.05–0.01, $P$ values of 0.01–0.001, or $P < 0.001$, respectively.

We next tested if TFEB nuclear mobilization was subject to temporal modulation by *Salmonella* grown to either log- or stationary-phases by analyzing the N/C ratio of TFEB at 1 or 4 h post-infection. As previously observed, macrophages infected with stationary-phase *Salmonella* displayed mostly cytosolic TFEB at 1 h post-infection and higher nuclear levels of TFEB at 4 h post-infection (Fig. 2A and C). In comparison, the N/C ratio of TFEB was similar between macrophages that engulfed log-phase *Salmonella* after 1 or

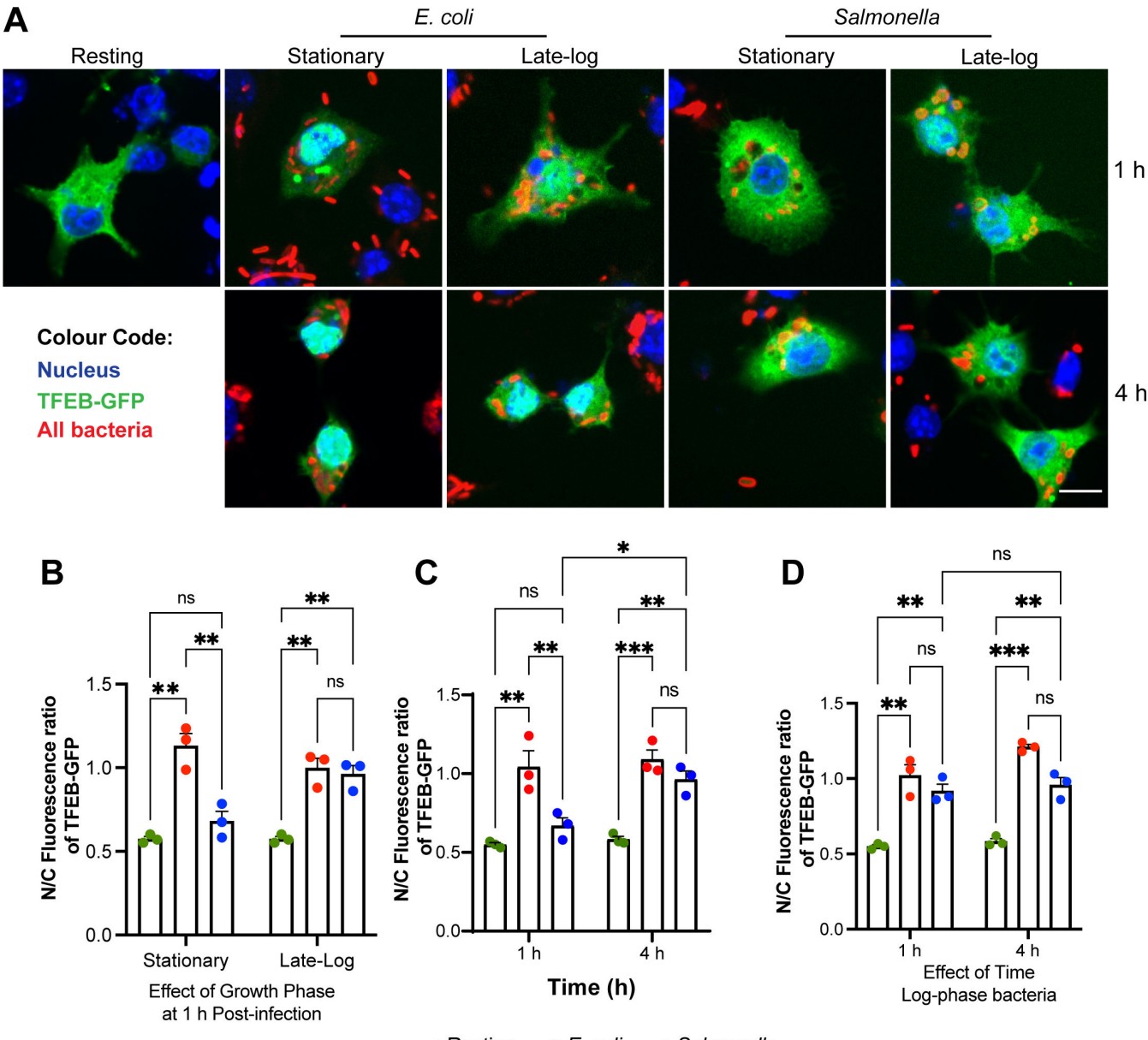

**FIG 2** Stationary-grown *Salmonella* but not log-phase *Salmonella* delays TFEB activation in macrophages. (A) Confocal micrographs of RAW cells expressing GFP-TFEB before, 1 h, or 4 h after engulfing living *E. coli* and *Salmonella* grown to stationary or late log-phase. After uptake, cells were fixed and stained with DAPI and with anti-bacteria antibodies to detect the nucleus and bacteria, respectively. (B) The N/C fluorescence ratio of GFP-TFEB in macrophages 1 h after engulfment of stationary- or late-log bacteria. (C) The N/C fluorescence ratio of GFP-TFEB in macrophages 1 or 4 h after engulfment of stationary-grown *E. coli* or *Salmonella*. (D) The N/C fluorescence ratio of GFP-TFEB in macrophages 1 or 4 h after engulfment of log-phase grown *E. coli* or *Salmonella*. For (B–D), data are shown as mean ± SEM from three independent experiments, scoring 70 cells for each condition per experiment. Means were tested using matched two-way ANOVA test and post hoc Tukey's test, where *, **, and *** indicate $P$ values of 0.05–0.01, $P$ values of 0.01–0.001, or $P < 0.001$, respectively.

4 h of infection, and both were statistically comparable to the counterpart macrophages that engulfed *E. coli* (Fig. 2A and D). Collectively, *Salmonella* modulates TFEB nuclear localization in macrophages in a manner dependent on its growth conditions and stage of infection. Specifically, *Salmonella* grown to stationary phase actively delays TFEB mobilization relative to *E. coli* or *Salmonella* grown to log-phase.

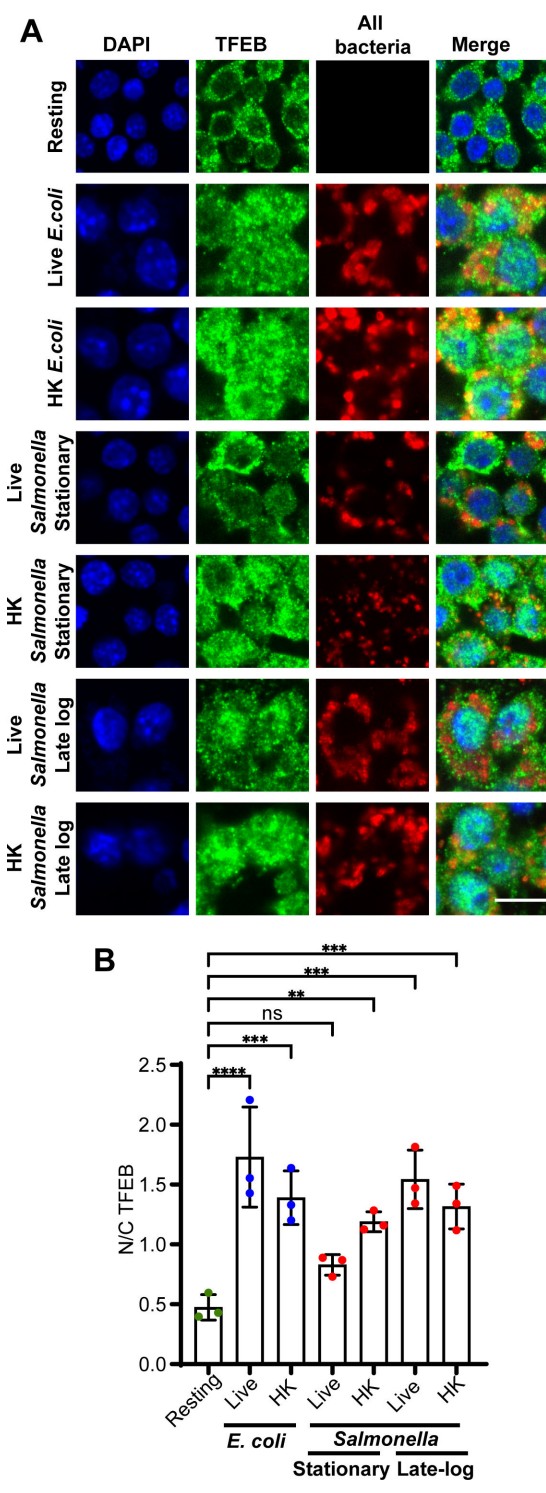

**FIG 3** The distribution of endogenous TFEB in macrophages after uptake of viable and non-viable *E. coli* and *Salmonella*. (A) Confocal micrographs of RAW macrophages incubated with living or heat-killed stationary-grown *E. coli* or living or heat-killed *Salmonella* grown to either stationary or late-logarithmic phase. After 1 h, cells were fixed and stained with DAPI (blue) to identify the nucleus, anti-TFEB antibodies (green), and anti-LPS antibodies to identify all bacteria (red). Scale bar = 10 µm. (B) N/C fluorescence ratio of endogenous TFEB in macrophages subject to indicated conditions. Data are shown as mean ± SEM from three independent experiments, scoring 80 cells for each condition and experiment. Means were tested using one-way ANOVA test with matched data and post hoc Dunnett's test, where *, **, ***, and **** indicate $P$ values of 0.05–0.01, 0.01–0.001, 0.001–0.0001, or $P < 0.0001$, respectively.

## *Salmonella* growth conditions affect TFEB nuclear translocation in HeLa cells

Given our observations in RAW macrophages above, we next assessed if *Salmonella* grown to either stationary or late-log also caused differential nuclear translocation of endogenous TFEB in a non-macrophage cell model, for which we used HeLa cells. First, endogenous TFEB remained cytosolic in HeLa cells infected with *Salmonella* grown to either growth phase after 1 h (Fig. 4A and B). Yet, after 4 h post-infection, we then observed an enrichment in nuclear TFEB relative to the cytosolic pool in HeLa cells infected with late-log *Salmonella*, but not in HeLa cells infected with stationary-grown *Salmonella* (Fig. 4A and B). The absence of nuclear TFEB in HeLa cells infected with stationary-grown Salmonella may reflect the less invasive nature of these bacteria, which could affect uptake by non-phagocytes. However, TFEB remained predominantly cytosolic even in HeLa cells with larger number of engulfed *Salmonella*. Overall, we reveal that TFEB nuclear translocation in non-macrophage cells is also dependent on *Salmonella* growth conditions, and hence, context-dependent.

## The effect of *Salmonella* on TFEB phosphorylation

The cytosolic and nuclear distribution of TFEB can be regulated by the phosphorylation status at various residues of TFEB (25, 26, 56–58). To better understand how *Salmonella* might affect TFEB distribution in macrophages, we used Phos-tag gels to determine the general phosphorylation of TFEB using whole cell lysates (59) and quantified the band migration front of TFEB. To analyze total levels, we employed a standard SDS-PAGE gel and Western blotting.

As expected, cell lysates from macrophages that engulfed *E. coli* for 1 h displayed a TFEB band pattern that migrated faster relative to lysates from resting, i.e., phagocytosis of *E. coli* caused a significant net dephosphorylation of TFEB (Fig. 5A and B). Interestingly, this was more variable in macrophages 4 h post-phagocytosis of *E. coli*. In comparison, infection with *Salmonella* for either 1 or 4 h appeared to have an intermediate and heterogeneous effect on TFEB phosphorylation (Fig. 5A and B). This result was not expected and may reflect a change in phosphorylation sites without a net change in number of phosphorylated residues of TFEB and/or heterogeneous infection rates that is not resolvable by a population-based assay such as Western blotting (58), or may suggest that TFEB is regulated through another mechanism such as the recently discovered itaconate modification of TFEB (37, 60). Interestingly, the levels of total TFEB normalized to actin were reduced in macrophages infected for 4 h with late-log phase *Salmonella*, indicating that under specific conditions, *Salmonella* can repress TFEB expression or promote its turnover (Fig. 5A, C, and D), consistent with *Salmonella* infection of murine bone marrow-derived primary macrophages (33). Overall, TFEB control by *Salmonella* is subject to a temporal sequence of events that include control of its levels and likely a complex pattern of post-translational modifications.

## T3SS-1 and T3SS-2 secretion systems are needed for stationary-phase grown *Salmonella* to delay TFEB nuclear mobilization

Our data show that stationary-phase *Salmonella* actively delays TFEB mobilization into the nucleus after macrophage engulfment. To test whether SPI-1 and SPI-2 T3SS were involved in this delay, we permitted macrophages to engulf stationary-phase and log-phase *Salmonella* carrying loss-of-function mutations in *invA* and *ssaR genes*, which are respectively required to express subunits and/or form the SPI-1 and SPI-2 complexes (41, 46, 61, 62). As before, wild-type *Salmonella* grown to stationary phase did not elicit nuclear accumulation of TFEB, unlike *Salmonella* grown to log-phase (Fig. 6A and B). However, macrophages that engulfed *invA ssaR Salmonella* grown to either growth conditions displayed the higher levels of nuclear TFEB relative to stationary-phase wild-type *Salmonella* (Fig. 6A and B). We note that single *invA* and *ssaR* mutants had intermediate effect on TFEB localization, suggesting some redundancy toward TFEB manipulation (not shown). Thus, *Salmonella* SPI-1/SPI-2 double mutants fail to repress

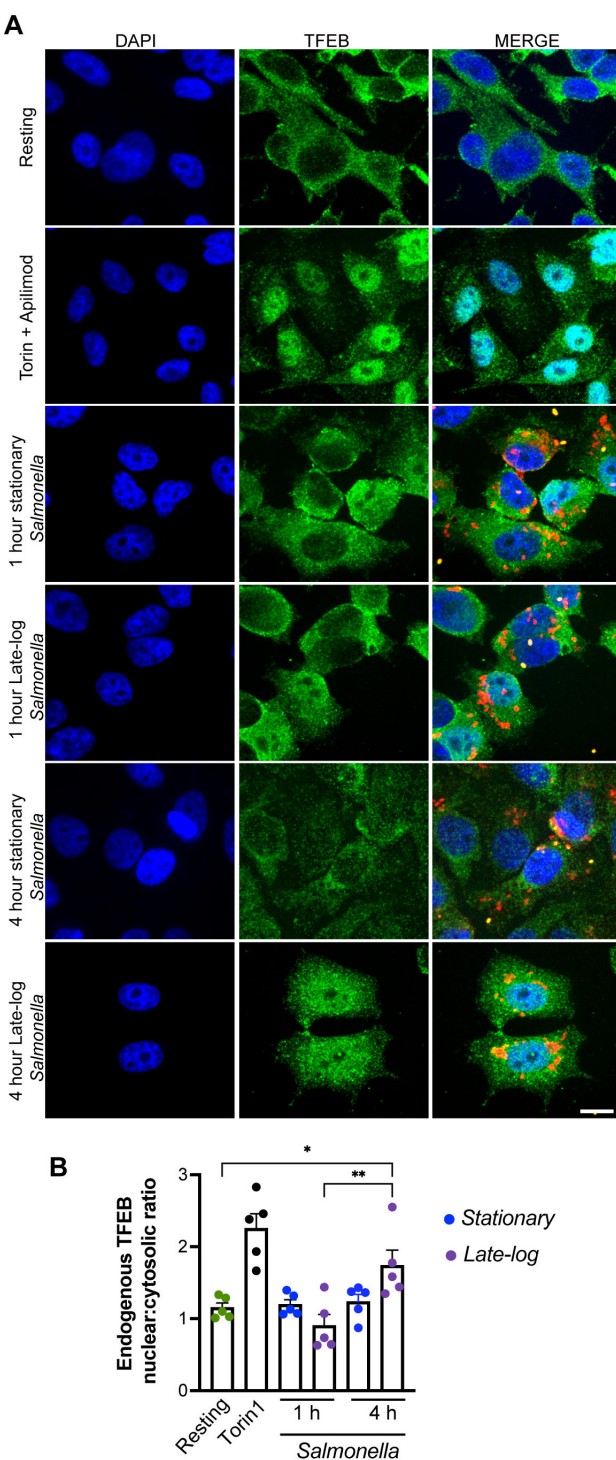

**FIG 4** TFEB distribution in HeLa cells after infection with *Salmonella*. (A) Confocal images of HeLa cells incubated with living stationary or late-log grown *Salmonella*. After 1 and 4 h post-infection, cells were fixed and stained for external *Salmonella* with anti-*Salmonella* serum and far-red secondary antibodies, followed by permeabilization and stained with DAPI (blue) to identify the nucleus, anti-TFEB antibody (green), and anti-*Salmonella* antiserum (red bacteria are internal; yellow are external). Images represent summed *Z* stack images. Scale bar = 10 µm. (B) N/C ratio of endogenous TFEB fluorescence in HeLa cells. Data shown represent mean values ± SEM. Statistical analysis was preformed using matched one-way ANOVA and post hoc Tukey's test, where * and ** indicate *P* values of 0.05–0.01 and 0.01–0.001, respectively.

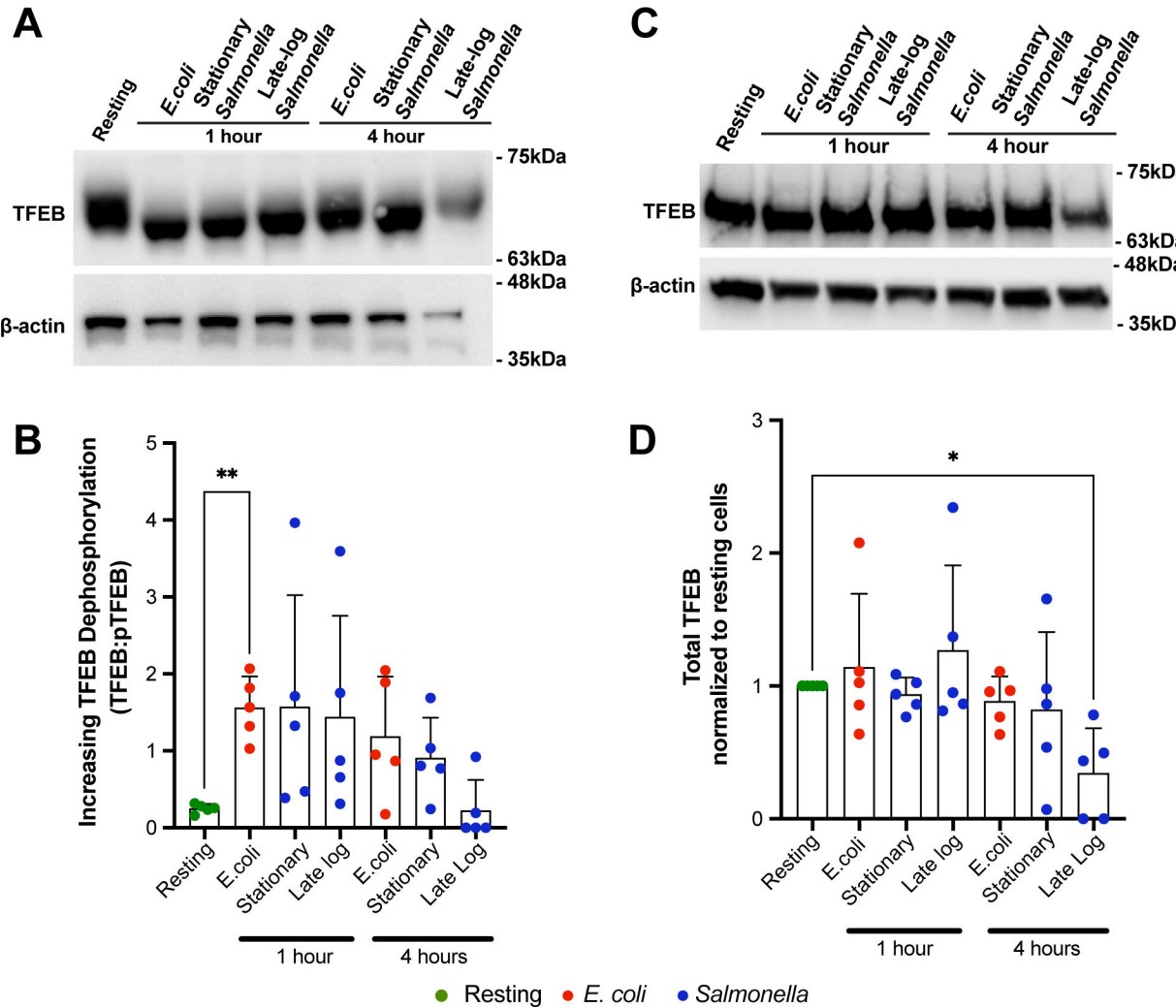

**FIG 5** TFEB phosphorylation and protein levels in macrophages after *Salmonella* infection. (A and C) Macrophages engulfed *E. coli* or *Salmonella* grown to stationary or log-phase for 1 or 4 h. Macrophages were then lysed and lysates separated on a Phos-tag SDS-PAGE (A) or standard SDS-PAGE (C), followed by blotting with anti-TFEB antibodies. The loading control was β-actin for both types of electrophoresis. (B and D). The migration front of TFEB separated in Phos-tag gels was measured as indicator of phosphorylation status of TFEB (B) and normalized total TFEB levels relative to β-actin (D). Shown is the mean ± SD analyzed from five independent experiments. Data in (B) were analyzed by a matched one-way ANOVA and Dunnett post hoc test, while data in (D) were tested by Friedman one-way ANOVA test and the Dunn's post hoc test. * and ** indicate a *P* value of <0.05 and <0.01, respectively.

TFEB activation during phagocytosis and we predict that TFEB activation is now a host-driven process. In conclusion, our data suggest that live *Salmonella* grown to stationary phase actively represses TFEB in a time-dependent and effector-dependent manner.

## TFEB repression depends on PhoP, SifA, and SopD2

Stationary-phase *Salmonella* appears to actively delay TFEB nuclear migration during macrophage infection. To further determine if stationary-phase *Salmonella* actively repress TFEB, we examined TFEB nuclear mobilization in macrophages after engulfment of seven *Salmonella* mutants; these mutants were either disrupted in the master virulence regulator, PhoP (Δ*phoP*) or eliminated effectors of the *Salmonella* SPI-2 system known to alter intracellular signaling and membrane trafficking (Δ*sopD2*, Δ*sopA*, Δ*sifA*, Δ*pipB2*, Δ*sspH2*, Δ*slrP*, or Δ*sigD*) (39, 63–65). These *Salmonella* mutants were grown to stationary phase since this is the bacterial growth condition that actively suppressed TFEB nuclear entry. Of these seven mutants, we identified three whose phagocytosis

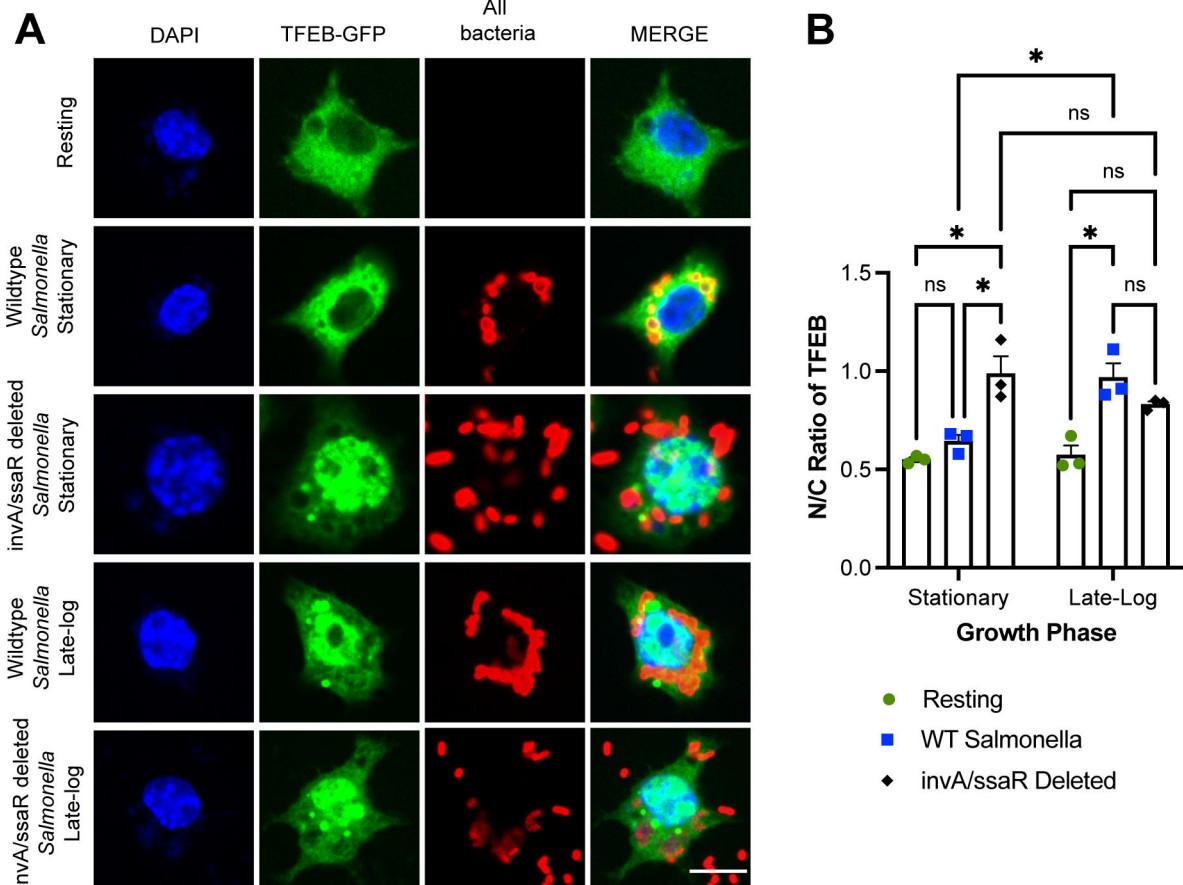

**FIG 6** SPI-I and SPI-II-deficient *Salmonella* do not stall TFEB activation in macrophages. (A) Confocal images of RAW cells expressing GFP-TFEB (green) 1 h after engulfing wild-type or *invA ssaR Salmonella* grown to stationary or late log-phase. After uptake, cells were fixed and stained with DAPI (blue) and with anti-bacteria antibodies (red). (B) The N/C fluorescence ratio of GFP-TFEB in macrophages after 1 h engulfment of *Salmonella* strains and grown to indicated phases. Data are shown as mean ± SEM from three independent experiments, scoring at least 70 cells for each condition per experiment. Means were tested using two-way ANOVA test and post hoc Tukey's test, where * and ** indicate *P* values of 0.05–0.01 and 0.01–0.001, respectively.

triggered significant activation of GFP-tagged and endogenous TFEB relative to wild-type *Salmonella*: *sopD2*, *sifA*, and *phoP* (Fig. 7). These data further support a functional role for *Salmonella* in impeding early TFEB activation after phagocytosis by macrophages. We speculate that the PhoPQ two-component system is necessary to sense the intra-phagosomal environment of engulfed *Salmonella* eliciting expression of effectors that delay TFEB activation such as effectors of the SPI-II system (49, 66–68). In turn, SopD2 and SifA are known to alter intracellular signaling and trafficking, which may prevent activation of TFEB (63, 69–71). These observations support a model whereby stationary-grown *Salmonella* actively repress TFEB, at least in the early stages of infection.

## Impact of *Salmonella* growth phase on gene expression in macrophages

To understand if the delay in TFEB nuclear entry by stationary-grown *Salmonella* reprogrammed lysosome and autophagy gene expression relative to infection with late-log *Salmonella*, we quantified by quantitative real-time PCR (qRT-PCR) the expression of the model genes, LC3, LAMP1, and cathepsin D. We used wild-type, *tfeb*$^{-/-}$, and *tfeb*$^{-/-}$ *tfe3*$^{-/-}$ RAW cells, previously described (17). First, deletion of TFEB and/or TFE3 did not significantly decrease basal expression of these genes (Fig. 8A). Second, the mRNA levels of LAMP1, cathepsin D, and LC3 all behaved distinctly depending on infection, time, and genotype. For example, neither genotype, infection mode, nor post-infection time

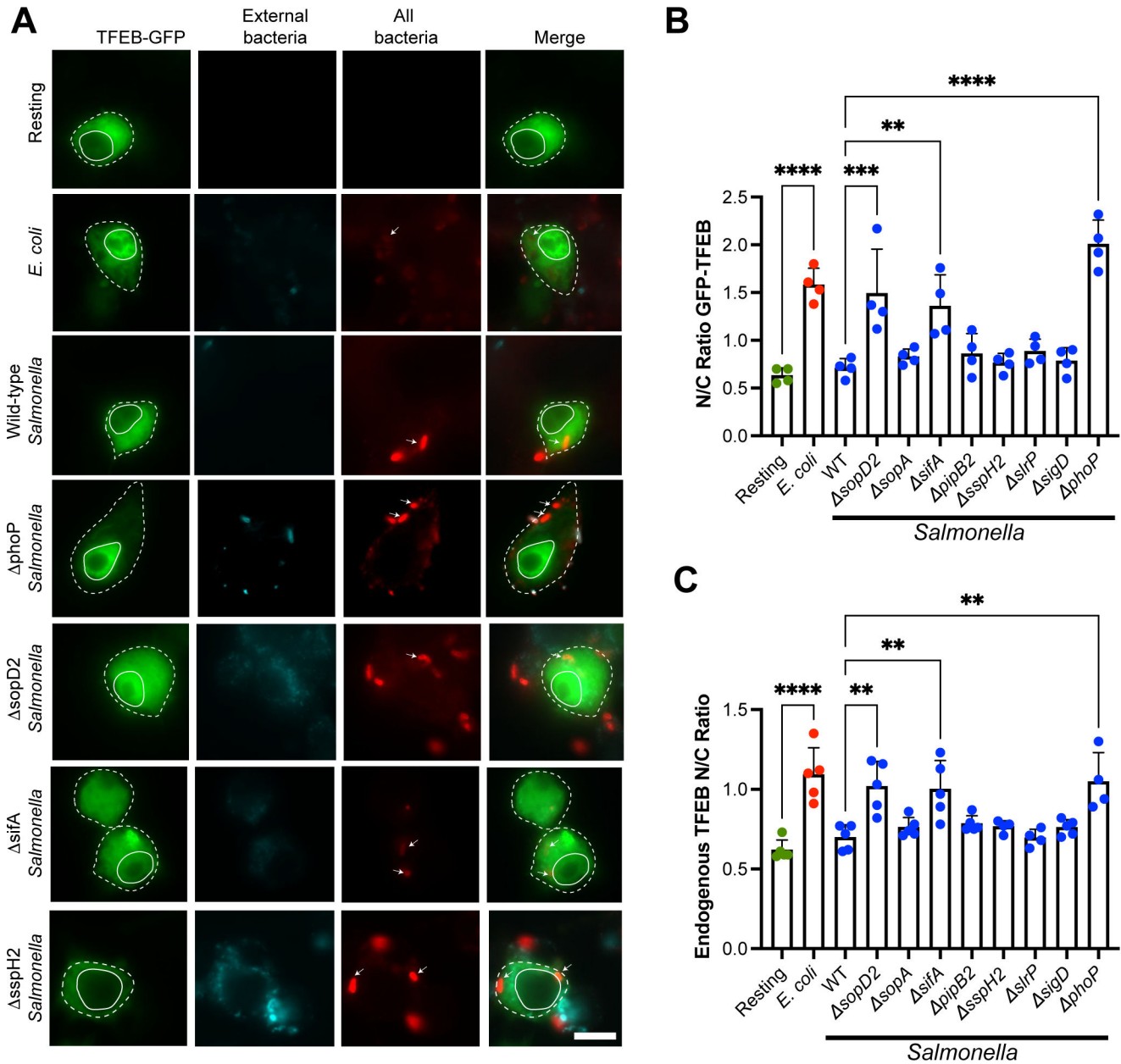

**FIG 7** Identification of *Salmonella* mutants that cannot suppress TFEB nuclear translocation in RAW macrophages. (A) Epifluorescence micrographs of RAW cells expressing GFP-TFEB 1 h after engulfment of indicated bacteria strains. Cells were fixed and stained for external (cyan) and internal bacteria (red) bacteria with anti-*E. coli* or anti-*Salmonella* antibodies. Arrows point to specific internal bacteria, while full and dotted outlines indicate nucleus and individual cell boundaries. Scale bar represents 10 µm. (B) N/C ratio of GFP-TFEB in macrophages after engulfment of indicated bacterial strains. (C) N/C ratio of endogenous TFEB in macrophages after engulfment of indicated bacterial strains. For (B and C), data are shown as mean ± SEM from four independent experiments with approximately 50 cells counted per condition per experiment. Means were tested using one-way ANOVA test and post hoc Tukey's test, where *, **, and *** indicate *P* values of 0.05–0.01, *P* values of 0.01–0.001, or *P* < 0.001, respectively.

significantly altered the levels of LAMP1 mRNA (Fig. 8B and C). In comparison, cathepsin D mRNA levels increased in wild-type RAW cells 20 h post-infection with either stationary or late-log *Salmonella* (Fig. 8D). However, it was unclear if this depended on TFEB/TFE3; while the increase in cathepsin D mRNA is eliminated in *tfeb*−/− macrophages after infection, the baseline appears higher in all conditions relative to wild-type macrophages (even if not statistically significant in Fig. 8A). Additionally, while stationary-grown *Salmonella* did not elicit a statistically significant increase in cathepsin D mRNA in *tfeb*−/−

*tfe3*⁻/⁻, unlike in wild-type macrophages, there was still an increase in cathepsin D mRNA in macrophages infected with late-log *Salmonella* (Fig. 8D). Moreover, we could not detect statistical differences when comparing the genotypes at 20 h post-infection (Fig. 8E). Finally, LC3 mRNA levels did not change in wild-type macrophages infected with either growth phases of *Salmonella* (Fig. 8F). Yet, and surprisingly, *tfeb*⁻/⁻ and *tfeb*⁻/⁻ *tfe3*⁻/⁻ macrophages displayed augmented LC3 mRNA expression levels infected with late-log *Salmonella* (Fig. 8F and G). Overall, stationary versus late-log *Salmonella* may elicit distinct gene expression patterns in infected macrophages and some of these may depend on TFEB/TFE3 but dissecting this may require more sophisticated approaches such as RNA-seq even single-cell transcriptomics given the variability in the number of bacteria per macrophage, which could affect TFEB function.

## Stationary and late-log *Salmonella* differentially impact autophagy in macrophages

We also queried the state of autophagy and xenophagy, both of which can be affected by *Salmonella* and TFEB. To this end, we transfected wild-type, *tfeb*⁻/⁻, and *tfeb*⁻/⁻ *tfe3*⁻/⁻ RAW cells with the mCherry-GFP-LC3 reporter. This chimeric protein reports on both the number of autophagosomes (mCherry puncta) and autophagosome maturation, which can be determined by the quenching of GFP fluorescence in the low pH of autolysosomes (72, 73).

First, we observed minor differences in the number of autophagosomes (LC3-mCherry) between wild-type, *tfeb*⁻/⁻, and *tfeb*⁻/⁻ *tfe3*⁻/⁻ RAW cells (Fig. 9A–D). Interestingly, wild-type RAW cells infected with late-log *Salmonella* generally had significantly fewer autophagosomes than macrophages infected with stationary-grown *Salmonella* (Fig. 9E). However, this difference between infection with stationary or late-log *Salmonella* remained when the host cells were mutated for TFEB and/or TFE3 (Fig. 9F and G). Moreover, we did not observe a significant difference in autophagosome maturation or xenophagy that depended on genotype of macrophages, bacterial growth phase, and time of infection (Fig. 10). Overall, again, while there may be differences on autophagy triggered by bacterial growth phase, our data does not indicate that these depend on TFEB manipulation, though it is possible that extending the time of infection may lead to differences based on trends observed.

## TFEB expression impacts *E. coli* and *Salmonella* survival differently in macrophages

Our observations indicate that TFEB distribution in macrophages during *Salmonella* uptake is dependent on *Salmonella* growth conditions, wherein stationary *Salmonella* transiently, but actively represses TFEB, while invasive *Salmonella* permits or elicits nuclear translocation of TFEB. In comparison, phagocytosis of non-pathogenic *E. coli* always triggers nuclear entry of TFEB in macrophages irrespective of the growth conditions of *E. coli* and over the observed period. This raises the question of whether TFEB helps repress *Salmonella* or enables *Salmonella* growth in macrophages.

To better understand this potential conundrum, we investigated the survival of stationary-grown *E. coli* and *Salmonella* within wild-type and *tfeb*⁻/⁻ RAW macrophages (17). To do this, we allowed macrophages to phagocytose bacteria for 1 h, followed by gentamicin treatment to kill external bacteria. Macrophages were then lysed or chased for an additional 4 h to, respectively, assess bacterial uptake and survival within macrophages. Lysates were plated and the number of colonies formed were scored and normalized against the initial uptake. For *E. coli*, we observed a significant increase in bacterial survival in *tfeb*⁻/⁻ RAW macrophages compared to wild-type RAW cells (Fig. 11A), consistent with previous results (18). By contrast, we did not observe a difference in *Salmonella* survival infecting wild-type or *tfeb*⁻/⁻ RAW macrophages (Fig. 11B). We then tested if the growth-phase of Salmonella affected survival in wild-type and in *tfeb*⁻/⁻ *tfe3*⁻/⁻ RAW cells (17). As before, we did not observe any significant differences in *Salmonella*

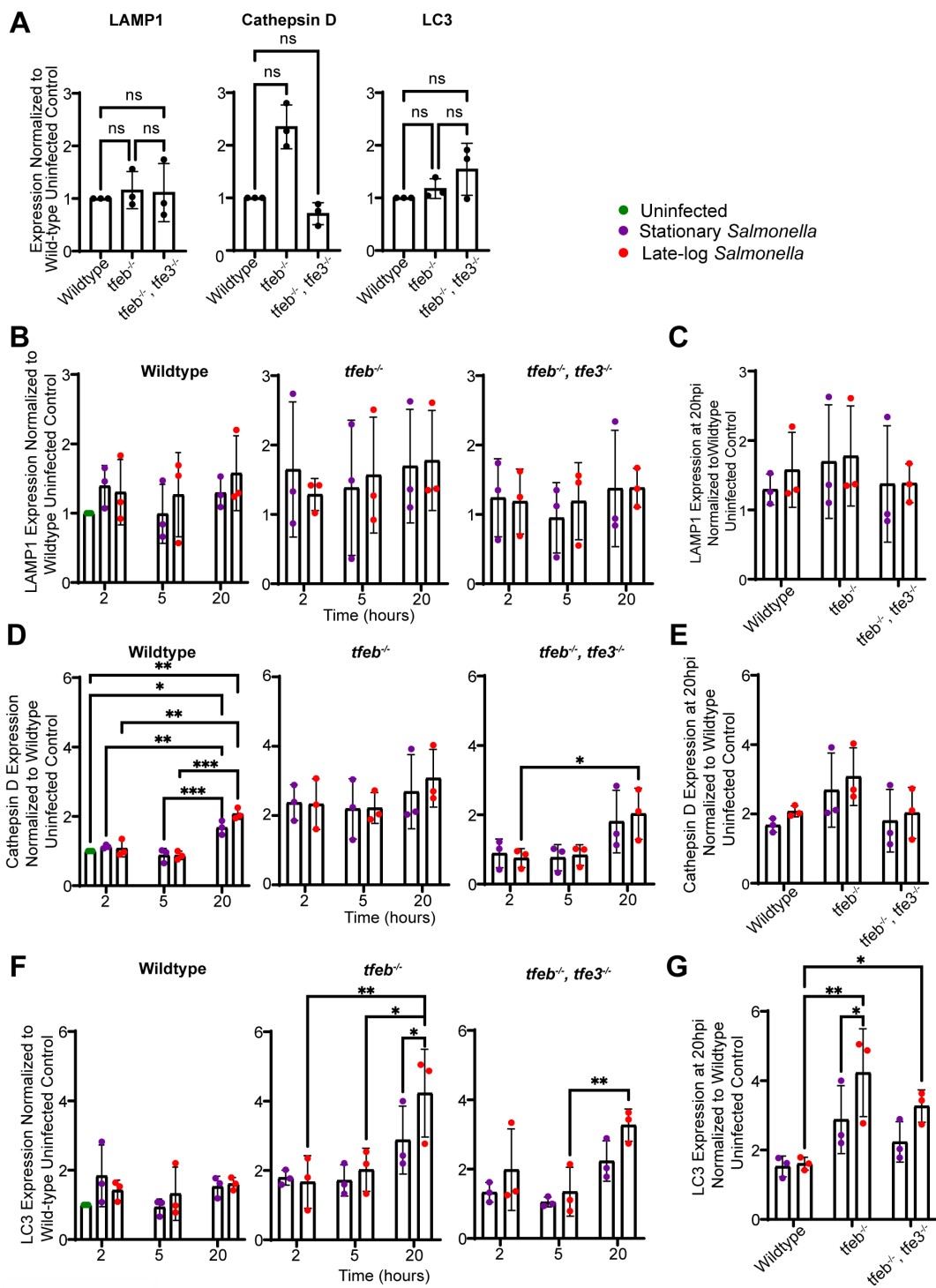

**FIG 8** mRNA levels of LAMP1, cathepsin D, and LC3 during *Salmonella* infection. (A) Basal expression of LAMP1, cathepsin D, and LC3 genes across wildtype, *tfeb*$^{-/-}$, and *tfeb*$^{-/-}$ *tfe3*$^{-/-}$ RAW macrophages normalized to wildtype. (B, D, and F) Relative LAMP1 (B), cathepsin D (D), and LC3 (F) mRNA expression within each cell type infected with either stationary or late-log grown *Salmonella* at indicated times post-infection. (C, E, and G) Comparison of LAMP1 (C), cathepsin D (E), and LC3 (G) mRNA expression at 20 h post-infection between wild-type, *tfeb*$^{-/-}$, and *tfeb*$^{-/-}$ *tfe3*$^{-/-}$. Data in (C, E, and G) are the same as 20 h from (B, D, and F). Data plotted represent means of individual replicates ±SD. Statistical analysis was preformed using matched one-way ANOVA and post hoc Tukey's test, where *, **, and *** indicate *P* values of 0.05–0.01, 0.01–0.001, and 0.001–0.0001, respectively.

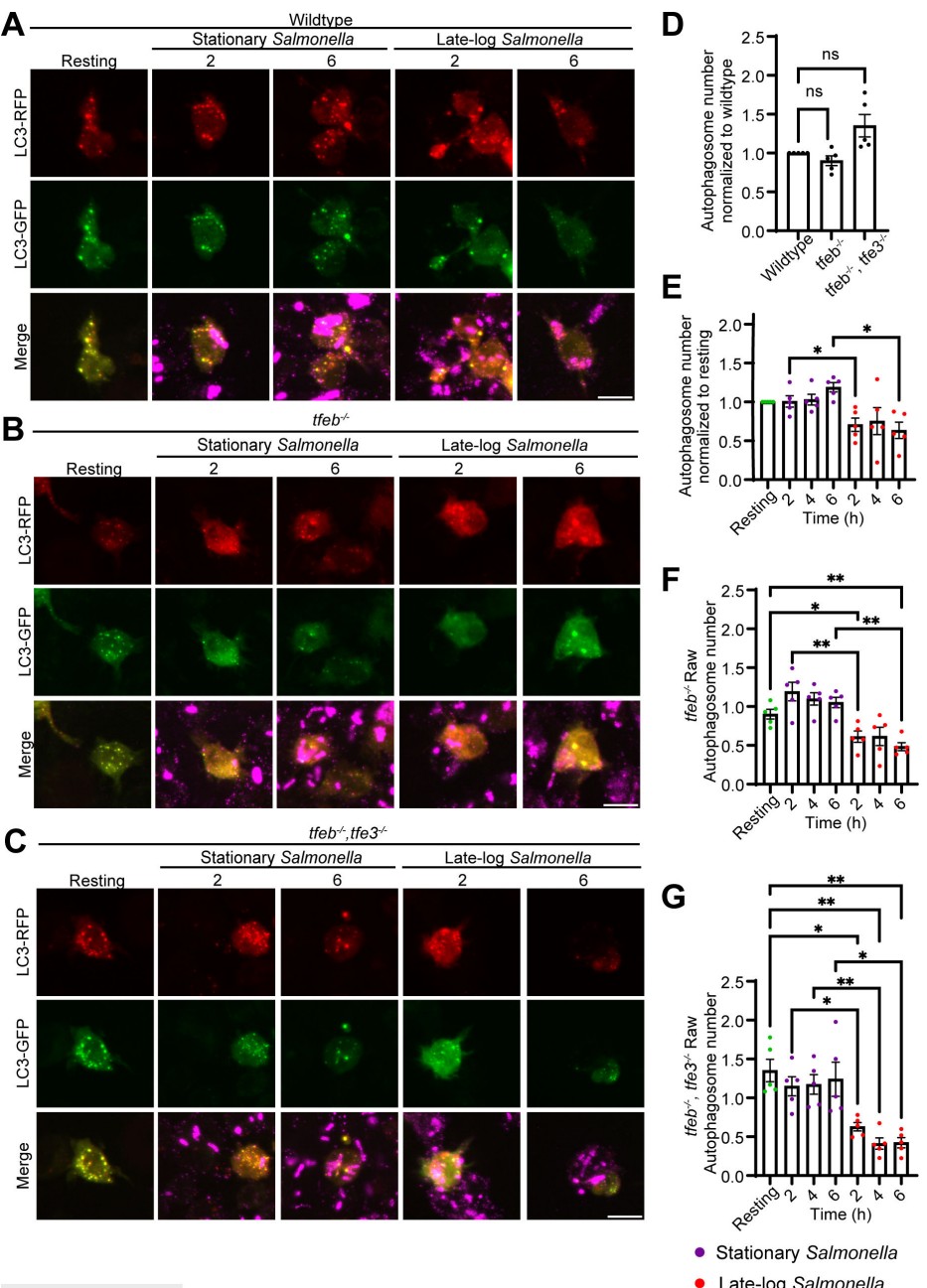

**FIG 9** Autophagy in RAW macrophages during stationary and late-log *Salmonella* infection. (A, B, and C) Confocal images of wildtype (A), *tfeb*⁻/⁻ (B), and *tfeb*⁻/⁻/*tfe3*⁻/⁻ (C)RAW macrophages transfected with mCherry-GFP-LC3. Cells were subject to stationary or late-log *Salmonella* infection and were fixed at 2, 4, and 6 h post-infection. Cells were also immunofluorescently stained for *Salmonella* using an anti-*Salmonella* antiserum. Panel shows 2 and 6 h post-infection images. Scale bar = 12 μm. (D) Quantification of autophagosomes in resting wildtype, *tfeb*⁻/⁻, and *tfeb*⁻/⁻/*tfe3*⁻/⁻ RAW macrophage. (E, F and G) Comparison of autophagosome numbers between stationary and late-log *Salmonella* infected macrophages over time for each respective cell type. *tfeb*⁻/⁻ and *tfeb*⁻/⁻/*tfe3*⁻/⁻ values are normalized against wild-type resting condition. Plotted values represent mean values per replicate and ±SEM. For (D), statistical analysis was done using one-way ANOVA and Friedman's multiple comparisons. Statistical analysis for (E and G) was preformed using matched two-way ANOVA and post hoc Tukey's test, where *, **, and *** indicate *P* values of 0.05–0.01, 0.01–0.001, and 0.001–0.0001, respectively.

survival at 4 h post-infection that were dependent on TFEB/TFE3 and growth-phase of the bacteria (Fig. 11C). At 20 h post-infection, we also did not observe a statistically significant difference between these conditions, but we note the low *P* values, which

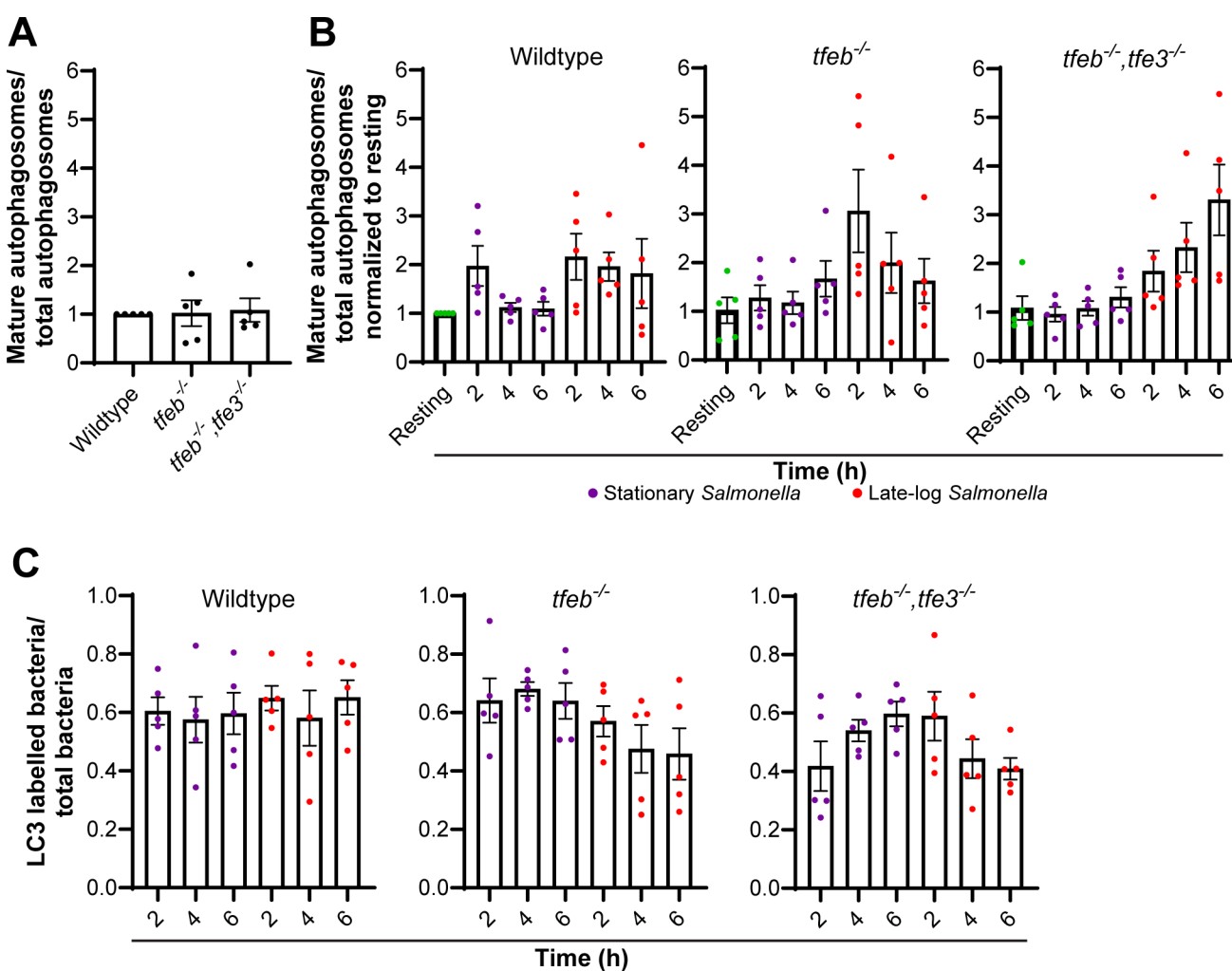

**FIG 10** Autophagosome maturation and xenophagy bacterial capture in RAW macrophages during Salmonella infection. (A) Ratio of mature over total autophagosomes in resting wildtype, *tfeb*$^{-/-}$, and *tfeb*$^{-/-}$/*tfe3*$^{-/-}$ RAW macrophages normalized to wildtype. (B) Ratio of mature to total autophagosomes during stationary or late-log infection in wildtype, *tfeb*$^{-/-}$, and *tfeb*$^{-/-}$/*tfe3*$^{-/-}$ RAW macrophages. (C) Ratio of LC3-labeled bacteria over total bacteria during infection in each macrophage genotype. Data plotted represent mean values and ±SEM from $n = 5$ independent experiments. Statistical analysis was preformed using matched one-way ANOVA and post hoc Tukey's test.

might indicate a trend (Fig. 11C). Overall, our observations suggest that TFEB aids macrophages in killing *E. coli*, but there is an equivocal effect on *Salmonella* survival, which may reflect a dual role of TFEB toward *Salmonella* that depends on its tight regulation by both host and microbe.

To complement our work in macrophages, we also investigated the importance of TFEB and related factors in intracellular survival of stationary and late-log grown *Salmonella* in HeLa cells. We used wild-type HeLa and HeLa cells triple-deleted for *tfeb*$^{-/-}$ *tfe3*$^{-/-}$ *mitf*$^{-/-}$ (*HeLa-TKO*; (29)) and measured *Salmonella* survival at 4 and 20 h post-infection. During the first 4 h post-infection, we did not observe a difference in *Salmonella* survival by either bacterial growth condition or host genotype (Fig. 11D). In contrast, we observed a significant difference in survival at 20 h post-infection, whereas stationary-grown *Salmonella* exhibited similar survival in wild-type and HeLa-TKO cells, late-log *Salmonella* coped better within HeLa-TKO cells relative to wild-type HeLa host cells (Fig. 11D). This indicates that TFEB-family proteins help neutralize *Salmonella* in non-macrophage cells.

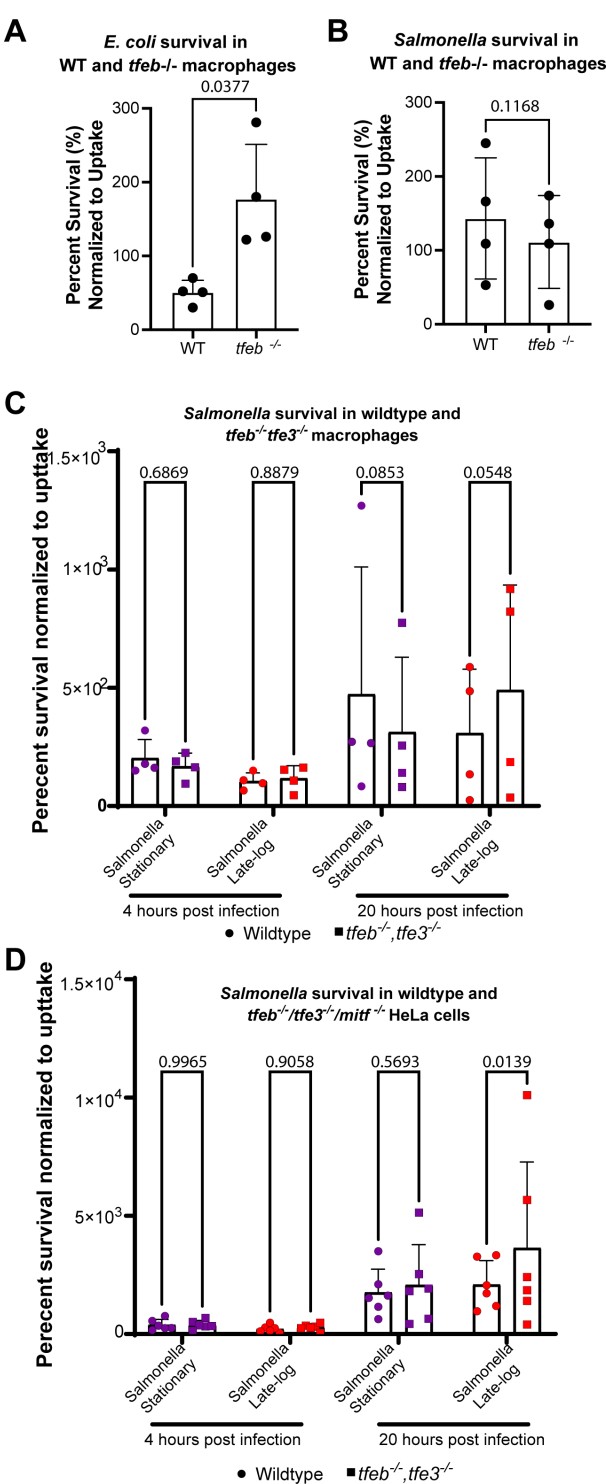

**FIG 11** *E. coli* and *Salmonella* survival in wild-type, *tfeb*$^{-/-}$ macrophages, and HeLA-TKO cells. (A and B) Wild-type and *tfeb*$^{-/-}$ RAW macrophages were allowed to phagocytose live *E. coli* (A) or *Salmonella* (B) for 1 h, followed with gentamycin to kill external bacteria. One macrophage population was immediately lysed and bacteria were plated to count CFUs as an estimate of bacterial uptake. A second population of macrophages was incubated for 4 h with low gentamycin, then lysed, and bacteria plated to count CFUs as indicator of acute bacterial survival in macrophages. Data are shown as percent survival by taking the ratio of CFUs after 4 h of incubation relative to CFUs formed after 1 h incubation (uptake). (C) Wild-type and *tfeb*$^{-/-}$ *tfe3*$^{-/-}$ RAW macrophages were allowed to engulf stationary and late-logarithmic

**FIG 11** (Continued)

grown *Salmonella*. (D) Infection of wild-type HeLa and *tfeb*$^{-/-}$ *tfe3*$^{-/-}$ *mitf*$^{-/-}$ HeLa (HeLa-TKO) cells as described in (C). Infections were followed for 4 and 20 h pos-infection. Percent survival was determined as described above. Data are shown as the mean ± SD from four (C) or six (D) independent experiments. Data in (A and B) were statistically analyzed by paired Student's *t* test, while data in (C and D) were tested by matched two-way ANOVA and Šidák post hoc test. *P* values are disclosed for survival assays.

## Pre-activation of TFEB by previous IgG-phagocytosis boosts *Salmonella* killing in macrophages

Our work suggests that *Salmonella* may orchestrate a temporal sequence of TFEB repression and activation that suits the stage of infection within macrophages. We hypothesized that disrupting this timed sequence may be detrimental to *Salmonella*. To test this, we employed an initial round of phagocytosis of IgG-coated beads, which we previously showed activates TFEB and enhances the degradation of lysosomes and killing of *E. coli* (18). Specifically, Fcγ receptor-mediated phagocytosis activates TFEB by releasing Ca$^{2+}$ from lysosomes during phagosome maturation, enhancing lysosomal gene expression (18). Thus, we pre-treated wild-type and *tfeb*$^{-/-}$ RAW macrophages with vehicle or IgG-coated beads, followed by phagocytosis of stationary phase *E. coli* or *Salmonella*, and then chased for 1 h to determine uptake or for 4 h to determine killing. The number of colonies formed was normalized to 1 h condition. As before, we observed increased *E. coli* survival in *tfeb*$^{-/-}$ macrophages relative to wild-type macrophages without priming (Fig. 12A; percent survival: 84 ± 5%, and 130 ± 11% in wild-type and in *tfeb*$^{-/-}$ macrophages, respectively). In comparison, *Salmonella* survived about the same in both genotypes (Fig. 12B; 160 ± 15% vs 187% ± 19% in wild-type and in *tfeb*$^{-/-}$ macrophages, respectively). When wild-type macrophages were primed with IgG-coated beads, we observed reduced survival of 57 ± 9% for *E. coli*, compared to 84% in unprimed wild-type macrophages (Fig. 12A). This drop in survival was eliminated in *tfeb*$^{-/-}$ macrophages (Fig. 12A; 130 ± 11% vs 137 ± 13%, unprimed vs primed). Interestingly, priming wild-type macrophages abated *Salmonella* survival (Fig. 12B; 94 ± 11% vs 160 ± 15%, primed vs unprimed), which was lost in *tfeb*$^{-/-}$ macrophages (Fig. 12B; 187 ± 19% vs 196 ± 26%, unprimed vs primed).

To complement these observations, we extended pre-activation of TFEB using apilimod, an inhibitor of PIKfyve that stimulates TFEB (74). However, apilimod treatment had little effect on *Salmonella* survival in both wild-type and *tfeb*$^{-/-}$ macrophages under our experimental conditions, which may reflect a more complex relationship between TFEB activation and PIKfyve-mediated membrane trafficking (Fig. 12C). Collectively, our observations with IgG-primed phagocytosis suggests that pre- or forced activation of TFEB can restrict *Salmonella* survival in macrophages.

## DISCUSSION

Recent work has highlighted the role of TFEB as a conserved immunoprotective transcription factor that promotes lysosomal and autophagic activities, antimicrobial peptide production, and modulation of cytokine secretion in response to various immune stimuli (17–19, 22, 24). Previous research showed that *Salmonella* can actively stimulate TFEB in RAW macrophages (22, 30), suggesting that TFEB activation provides an advantage to this microbe. However, this model appears inconsistent with observations that (i) *Salmonella* represses TFEB to escape macrophage clearance and (ii) acacetin-induced activation of TFEB in HeLa cells enhances the killing of *Salmonella* by stimulating xenophagy (33, 34). Collectively, this implies that the relationship between *Salmonella* and TFEB is multi-layered and possibly dependent on infection conditions. Indeed, our observations support this view since the effect of *Salmonella* on TFEB was affected by the bacterial growth conditions and time of infection, whereby stationary, but not late-log bacteria, delayed TFEB activation. Furthermore, inactivation of the SPI-1 and SPI-2 T3SS of *S.* Typhimurium abrogated the inhibition of TFEB-nuclear translocation. The same

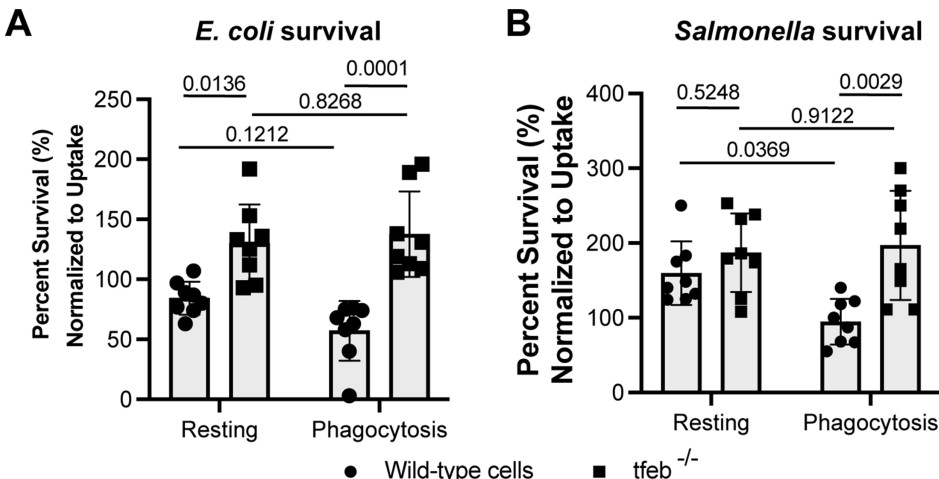

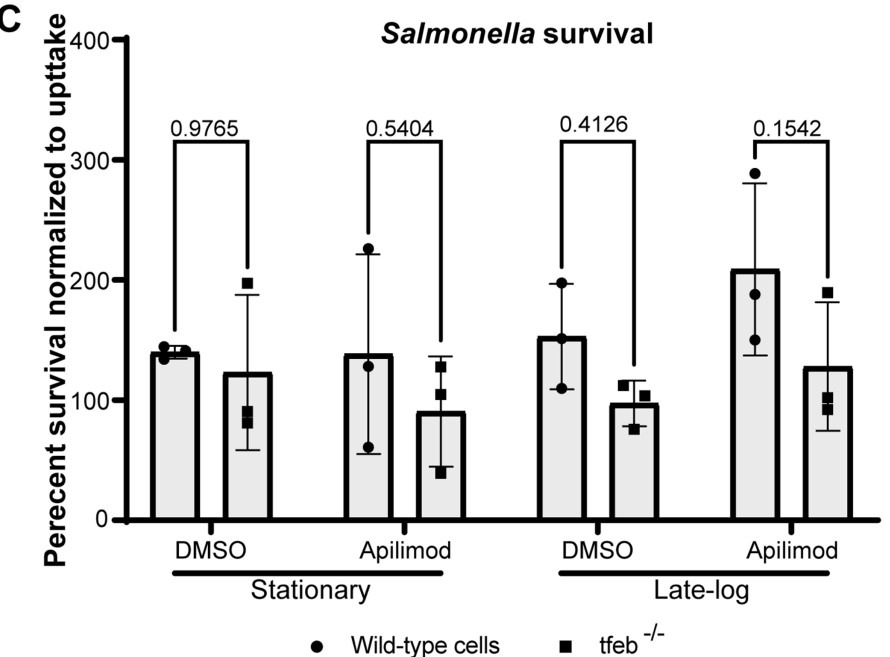

**FIG 12** Ectopic activation of TFEB enhances *Salmonella* killing in macrophages in a TFEB-dependent manner. (A and B) Wild-type and *tfeb*[−/−] RAW macrophages were left untreated or pre-activated with IgG-opsonized beads as described in Materials and Methods and then allowed to engulf live *E. coli* (A) or *Salmonella* (B). Percent survival of *E. coli* (A) or *Salmonella* (B) was determined by comparing number of CFUs between initial uptake and 3 h post uptake. Shown is the average ± STD from *n* = 8 independent experiments. (C) Cells were pre-treated with 20 nM apilimod or DMSO vehicle for 1 h and maintained through the infection with stationary or late-logarithmic *Salmonella*. Percent survival of bacteria was measured by comparing the number of CFU at the end of 4 h chase with CFUs from initial uptake (1 h) as described in Materials and Methods. Shown is the mean and standard deviation from *n* = 3 independent experiments. (A–C) Data were analyzed by matched two-way ANOVA and Šidák's post hoc test. Actual *P* values are shown for survival experiments

effect was also seen when the master regulator, *phoP*, or the secreted effectors *sopD2* or *sifA* were deleted. Moreover, while deleting TFEB in macrophages protected *E. coli* from phagocytic killing, there was no effect on *Salmonella* survival. In comparison, promoting TFEB before infection enhanced the killing of both bacterial species.

## Host cell-dependent upregulation of TFEB

TFEB is subject to a complex regulatory network involving phosphorylation and other post-translational modifications (37, 58, 75). In the context of phagocytosis, host cells engage a signaling axis that promotes reactive oxygen species (ROS), which activates CD38 and synthesis of the second messenger NAADP (22). This in turn seems to activate MCOLN1/TRPML1 to release intraphagosomal $Ca^{2+}$ to stimulate calcineurin, which then dephosphorylates and promotes TFEB nuclear entry (18, 22). This process is clearly driven by the host cell in response to microbe engulfment and phagocytosis (18, 22). Additionally, host macrophages engage a TFEB-Irg1-itaconate signaling axis in response to *Salmonella,* whereby TFEB promotes Irg1-driven itaconate production. Itaconate then alkylates and promotes TFEB nuclear translocation, which helps restrict *Salmonella* survival (37, 60). As described, the activation of TFEB is macrophage driven. We propose that stationary *Salmonella* actively delays macrophage-driven TFEB stimulation because living stationary *Salmonella* repressed TFEB nuclear translocation, while non-viable stationary *Salmonella* or mutants in *invA ssrA, phoP, sifA,* and *sopD2* did not. We do not know how stationary *Salmonella* delays TFEB activation, but we propose here that virulence factors like SopD2 may prevent NAADP-$Ca^{2+}$ signaling or block itaconate action. In fact, *Salmonella*, as well as *Yersinia pestis* can degrade itaconate to enhance their survival in macrophages since itaconate compromises the glyoxylate cycle whereby bacteria can use host cell resources for energy production (76, 77). Finally, similar mechanisms may be at play during infection of non-macrophages since stationary *Salmonella* did not elicit TFEB nuclear entry in HeLa cells, while infection with late-log *Salmonella* did after 4 h.

## Salmonella-dependent upregulation of TFEB

Viable *Salmonella* was observed to more readily promote TFEB compared to non-viable *Salmonella* by engaging a conserved PC-PLC-PKD pathway (22, 30). This contrasts with observations that *Salmonella* represses and reduces TFEB levels (33). In our hands, while stationary *Salmonella* actively delays TFEB nuclear translocation, we did observe that late logarithmic-phase *Salmonella* triggers TFEB early during infection, which is consistent with Najibi et al. (30). However, contrary to their observations, dead *Salmonella* still activated TFEB in RAW macrophages—we do not know the reason for this discrepancy. It is also not unknown if any benefit exists for late-log *Salmonella* to promote or permit TFEB engagement; for example, it may be that specific temporal activation of TFEB may aid *Salmonella* in remodeling phagosomes or *Salmonella*-containing vacuoles. Perhaps aligning with the idea that the stage of infection is critical for what TFEB might do, we observed that late-log *Salmonella* reduced TFEB levels after 4 h of infection, consistent with reduced TFEB expression and lysosomal activity observed by Rao et al. (33).

## Mechanistic control of TFEB by *Salmonella*

How might stationary-phase *Salmonella* delay TFEB activation? Control of the expression of the SPI-1 and SPI-2 T3SS and their effectors depends on numerous regulatory inputs, including the phase of *Salmonella* growth (38–40). Specifically, late logarithmic phase growth promotes the expression of SPI-1 and epithelial cell invasion, while stationary phase-growth promotes SPI-2 expression (50, 53, 54). We note that single mutants SPI-1 (*invA*) and SPI-2 (*ssaR*) had intermediate effects on TFEB nuclear entry relative to wild-type and *invA ssaR* strains (not shown)—this may represent redundancy between T3SSs relative to TFEB modulation or adaptation of the single mutant bacteria by upregulating other T3SSs. On the other hand, *invA ssaR* double mutants grown to stationary phase failed to delay TFEB nuclear translocation, suggesting that *Salmonella* effectors are required for this. Given that *phoP, sifA,* and *sopD2* mutants also failed to delay TFEB nuclear translocation, this suggests that the PhoPQ two-component system and alteration of membrane and intracellular signaling are involved in delaying TFEB nuclear entry. SopD2 is proposed to be a GTPase activating protein for the Rab7 GTPase,

while SifA interacts with motor proteins to sculpt *Salmonella*-containing vacuoles into *Salmonella*-induced filaments (63, 69, 71).

We do not know how these factors interfere with TFEB, but we propose several non-mutually exclusive possibilities. First, BioID analysis of SopD2 and SifA identifies LAMTOR subunits as potential interactors (78), which may allow *Salmonella* to modulate mTORC1, responsible for phosphorylation and repression of TFEB. While we did not see a significant change in TFEB phosphorylation by Phos-tag gels in *Salmonella*-infected cells, this could reflect a different phosphorylation pattern without a change in net phosphate number. Second, stationary *Salmonella* may blunt ROS, thus preventing NAADP-$Ca^{2+}$ activation of TFEB after phagocytosis in macrophages (22). In fact, *Salmonella* appears to manipulate ROS in host cells (79, 80), though it is not clear if this depends on the growth phase and time of infection. Third, as indicated above, stationary *Salmonella* may blunt itaconate action, thus reducing TFEB activation (37, 60, 76, 77).

## Functional outcomes of *Salmonella* growth phase and TFEB regulation

Here, we reveal that infecting with stationary or late-log grown *Salmonella* has distinct effects on host cells. First, TFEB nuclear activation was delayed or repressed in macrophages and HeLa cells infected with stationary-grown *Salmonella*, but not with late-log *Salmonella*. Second, late-log *Salmonella* increased the levels of LC3 mRNA in TFEB and TFEB/TFE3-deleted macrophages relative to stationary-grown *Salmonella*. Third, and despite the second point above, late-log *Salmonella* reduced the number of LC3-positive autophagosomes relative to stationary-grown *Salmonella* in macrophages. However, changes in LC3 and cathepsin D mRNA and the number of autophagosomes cannot readily be assigned to TFEB modulation by the growth phase of the bacteria. Part of the challenge may be the heterogeneity among single *Salmonella* cells in a given population, how many bacteria infect a macrophage, and the time of infection. Resolving the role of TFEB delay by stationary-grown *Salmonella* may thus require single-cell RNA-seq analyses or single-cell, time-resolved assays of gene expression. Nevertheless, there is clearly a difference in host outcomes when infecting with stationary versus late-log *Salmonella*.

This may have implications for *in vivo* infection as well since exponential phase *Salmonella* are more readily cleared and killed by phagocytes in blood compared to stationary phase *Salmonella* (81). Stationary phase *Salmonella* also resulted in reduced colonization in livers and spleen and prevented animal death by 12–18 h compared to exponential phase bacteria suggesting greater virulence of exponential phase *Salmonella* as well as a greater ability to replicate *in vivo (*81). Interestingly, another study showed the transition of *Salmonella* from exponential to stationary phase can induce rapid and extensive apoptosis of infected macrophages but not epithelial cells; a process dependent on the invA gene, required for functional T3SS, and hilA gene, a regulator of SPI-1 gene expression (41). While these differences among stationary and late-log *Salmonella* may reflect alterations to the bacteria themselves, such as LPS remodeling (81, 82), modulation of host factors like TFEB may be important. Supporting this, we did observe an improved survivability of late-log *Salmonella* in HeLa cells deleted for TFEB and related factors relative to wild-type HeLa cells and to stationary *Salmonella*.

## Ectopic activation of TFEB enhances bacterial killing in macrophages

TFEB plays a protective role against intracellular bacteria (18, 24, 36, 37, 60). In fact, we previously showed that pre-activation of TFEB by stimulating Fcγ receptors with aggregated IgG, enhanced *E. coli* killing in macrophages. In this context, TFEB activation required phagosome-lysosome fusion and the release of lysosomal $Ca^{2+}$ via TRPML1 (18). More recently, pre-activation of TFEB by acacetin also promoted the killing of *Salmonella* and *S. aureus* (33, 34). This suggests that strategies that obligately activate TFEB, thus bypassing any spatiotemporal manipulation of TFEB by *Salmonella*, may be detrimental to this microbe. Indeed, we observed that phagocytosis of IgG-coated beads, which stimulates TFEB, enhanced killing of both *E. coli* and *Salmonella* in wild-type but not

in $tfeb^{-/-}$ RAW macrophages. Thus, while it may be that *Salmonella* benefits from TFEB activation under specific conditions, treatments that erode control of *Salmonella* on TFEB appear detrimental to the microbe.

Overall, our work sheds additional light on the interactions between *Salmonella* and TFEB in host macrophages. Depending on bacterial growth conditions and stage of infection, we find that *Salmonella* can either delay TFEB activation, or permit, or even elicit TFEB activation. This raises the possibility that TFEB presents a double-edged sword for *Salmonella*—depending on infection stages, TFEB may be detrimental or beneficial to *Salmonella*. Importantly, forcibly promoting TFEB can override the spatio-temporal control of TFEB imposed by *Salmonella*, enhancing the killing capacity of macrophages. Thus, therapeutics that promote TFEB and related proteins may be useful to enhance the killing of intracellular pathogens.

## MATERIALS AND METHODS

### Cell culture, plasmids, and transfection

The RAW 264.7 macrophage-like male murine cell line and the HeLa female human cell line were grown in Dulbecco's modified Eagle's medium (DMEM) supplemented with 10% (vol/vol) heat-inactivated fetal bovine serum (FBS) (Wisent, QC) at 37°C and 5% $CO_2$ up to passage 20. Before plating, cells were washed with sterile phosphate-buffered saline (PBS, Wisent) and released by scraping in 5 mL of DMEM media. Cells were seeded into 6-well or 12-well plates at a confluency of 40% and used the following day. Using the same growth conditions, we also used RAW 264.7 cells that were deleted for *TFEB* gene (RAW $tfeb^{-/-}$ cells) or double-deleted for *TFEB* and *TFE3* genes (RAW $tfeb^{-/-}$ $tfe3^{-/-}$ cells) using CRISPR-Cas9. As control cells, we used RAW cells subject to non-targeting CRISPR-Cas9 process (wild-type RAW cells). These cell lines were obtained from Dr. Rosa Puertollano (National Institutes of Health, MD) and were previously described (17). We also used HeLa cells deleted by CRISPR-Cas9 for TFEB, TFE3, and MITF (HeLa TKO), obtained as a kind gift from Dr. Richard Youle and previously characterized (29).

To express GFP-TFEB fusion protein, we used a plasmid provided by Dr. Shawn Ferguson (Yale University School of Medicine, CT) and previously described (26). A plasmid #22418 encoding mCherry-GFP-LC3 was obtained from Addgene, originally provided by Dr. Jayantha Debnath and its use was previously defined (72, 73). RAW cells were seeded at a 20–30% confluency in a 12-well plate the day before, and then transfected with the plasmid encoding GFP-TFEB or mCherry-GFP-LC3 using FuGene HD transfection reagent (Promega, WI), following the manufacturer's instructions. The transfection mix was left on cells for 5 h, followed by removal and addition of new media. Cells were used the following day.

### Bacterial strains and culture

In this study, we used *E. coli* DH5α, wild-type *Salmonella enterica sv.* Typhimurium (SL1344), and *S.* Typhimurium strain were deleted for the *phoP* gene (Δ*phoP*) (83). *S. typhimurium* mutants [Δ*sopD2*, Δ*sifA*, Δ*slrp*, Δ*sigD* (Δ*sopB*), Δ*sspH2*, Δ*sopA*, and Δ*pipB2*] were obtained from Dr. John H. Brumell at the Hospital for Sick Children (Toronto, ON). All bacterial strains were grown overnight in lysogeny broth (LB) at 37°C with shaking. To use stationary cultures, bacteria were pelleted, washed once with PBS, and resuspended in DMEM at $OD_{600}$ ~0.8 and immediately used for infection. To prepare log-phase growth bacteria, an overnight culture was sub-cultured in the morning at 1:10 in fresh LB and then grown at 37°C for 3.5 h until reaching late-logarithmic phase ($OD_{600}$ ~0.8), before pelleting, washing in PBS and resuspending in DMEM at $OD_{600}$ ~0.8.

### Phagocytosis and infection assays

For phagocytosis/invasion assays, RAW macrophages plated at $1–2 \times 10^6$ cells/well were exposed to either stationary-phase or late log-phase bacteria at a 100:1 multiplicity of

infection (MOI). Phagocytosis was synchronized through centrifugation of the culture plates at 500 × $g$ for 5 min at room temperature. Macrophages were allowed to phagocytose for 1, 4, or 6 h, unless stated otherwise. Depending on the application, cells were then subject to 50 µg/mL gentamicin to kill extracellular bacteria for 30 min, followed by either lysis with Laemmli buffer for Western blot analysis or fixation with 4% (vol/vol) paraformaldehyde (PFA) in PBS for 20 min and processing for imaging as described below. A similar approach was done for HeLa cell infections.

## Bacterial survival assays

Prior to engulfment of unopsonized *E. coli* or *S.* Typhimurium, RAW macrophages remained untreated or were pre-activated with IgG-opsonized beads for 1 h, followed by a 3 h chase. Treated or untreated macrophages were then infected with stationary-phase bacteria (*E. coli* and *S.* Typhimurium) at a final OD $_{600}$ = 1.0. Uptake was synchronized through centrifugation at 400 × $g$ for 5 min. At 1 h post-infection, macrophages were washed 3× with PBS and replaced with DMEM media containing 50 µg/mL gentamicin to kill extracellular bacteria for 30 min. After, cells were washed 3× with PBS and either lysed immediately (uptake) with 200 µL 1% Triton X-100 for 5 min or the media were replaced with new DMEM supplemented with 10% FBS, containing low concentration of gentamicin (12µg/mL), and incubated for 3h at 37°C to allow time for phagosome maturation and bacteria killing. Prior to lysing, cells were washed multiple times with PBS and replaced with DMEM media containing 50 µg/mL gentamicin to kill extracellular bacteria for 30 min. They were then washed 3× with PBS and lysed with 200 µL 1% Triton X-100 for 5 min and scraped to release bacteria. Cell lysates were resuspended in 800µL PBS and subjected to serial dilutions in LB broth, followed by plating of 10 µL in LB plates, and incubated overnight at 37°C. Colony numbers were counted and recorded to determine CFU/mL relative to uptake.

For *Salmonella* survival in wild-type and mutant RAW and HeLa cells, the above was done with some changes. Cells were plated in 12-well plates and used at a density of 1 × 10$^6$ cells/well and infected with *S.* Typhimurium grown to stationary or late-logarithmic phase at an MOI of 100:1. Cells were lysed immediately after 1 h of phagocytosis followed by 30 min incubation in DMEM with 50 µg/mL gentamicin to estimate bacterial uptake or lysed at 4 or 20 h post-infection to estimate killing. Lysates were then resuspended in 500µL of PBS, serially diluted and 10µL were plated on LB plates, and these were incubated overnight. Colony numbers were counted and used to determine bacterial survival relative to uptake.

For pre-activation of TFEB by apilimod, macrophages were treated with vehicle or 20 nM apilimod (Toronto Research Chemicals, Toronto, ON) for 1 h, followed by bacterial infection with the drugs present. Cells were then lysed immediately to determine uptake of bacteria or chased for 4 h post-infection with the drug to determine survival.

## Immunofluorescence and fluorescence microscopy

To visualize external bacteria in TFEB-GFP transfected cells, immunostaining was conducted by briefly washing cells 3× with 0.5% bovine serum albumin (BSA) and then incubating cells for 30 min with either rabbit anti-*E. coli* antibodies (Bio-Rad, ON) or rabbit anti-*Salmonella* O antiserum group B antibodies (Fisher Scientific, ON) at 1:500. Cells were then washed 3× with 0.5% BSA in PBS, followed by secondary donkey anti-rabbit antibodies conjugated to Dylight355 (Bethyl Laboratories, TX) for 30 min at 1:500 in 0.5% BSA in PBS. To visualize internal bacteria, cells were permeabilized with 1% Triton X-100 in PBS, followed by incubation with the same primary antibodies above, but followed with donkey anti-rabbit antibodies conjugated to Dylight 550 (Bethyl Laboratories, TX). Cells were washed 3× with 0.5% BSA and mounted onto glass slides with fluorescent mounting media (DAKO) for imaging.

To stain for endogenous TFEB, cells were fixed with 4% PFA (vol/vol) at room temperature for 20 min and 0.1 M glycine in PBS for 10 min, before being washed 3× with 0.5% BSA and permeabilized with 0.1% Triton for 10 min or using ice cold methanol

for 5 min. Cells were blocked for 30 min with 0.5% BSA in PBS, before incubating for 30 min with 1:250 dilution of rabbit anti-TFEB antibodies (Bethyl Laboratories, TX) in 0.5% BSA. Cells were then washed 3× with 0.5% BSA, incubated with 1:500 dilution of Dylight 488-conjugated donkey anti-rabbit antibodies (Bethyl Laboratories, TX) for 30 min. In some experiments, cells were also incubated for 5 min with 0.4 µg/mL 4′,6-diamidino-2-phenylindole (DAPI), washed, and mounted onto glass slides with fluorescent mounting media (DAKO) for imaging.

Fluorescently labeled samples were imaged with one of two microscopy systems. For epifluorescence, we used a 60× oil immersion objective mounted on an Olympus IX83 Inverted Microscope linked to a Hamamatsu ORCA-Flash 4.0 digital camera (Olympus, ON). Images were either used as is or subjected to deconvolution using CellSens Dimension Module (Olympus, ON) with the advanced maximum likelihood estimation algorithm. Alternatively, confocal imaging was done using a Quorum DisKovery spinning disc confocal microscope system equipped on a Leica DMi8 microscope, with a 63× 1.4 NA oil-immersion objective, and linked to an iXON 897 EMCCD camera. The system was controlled by the Quorum Wave FX powered by MetaMorph software (Quorum Technologies, Guelph, ON).

## mCherry-GFP-LC3 autophagy assay

After 24 h of transfection with mCherry-GFP-LC3 plasmid, RAW cells were subjected to an infection assay using both stationary and late-log grown *Salmonella* at an MOI of 100:1. Cells were then followed for 2, 4, and 6 h. At each time point, cells were fixed with 4% (vol/vol) PFA and processed for immunofluorescent staining of *Salmonella* as described above. Images were acquired using a spinning disk confocal microscope as above. To count total autophagosomes, mCherry-positive structures were counted in transfected cells. To count bacteria capture, total internal *Salmonella* in cells were scored and a ratio was formed between LC3-associated bacteria and total bacteria. To assess autophagosome maturation, the ratio of mCherry-only structures (no GFP) over total autophagosomes (all mCherry) was done. All conditions shown in Fig. 9 and 10 were acquired concurrently. To facilitate analyses and better manage comparisons, these data were split between different graphs.

## Image analysis

To quantify nuclear to cytosol TFEB intensity, images were acquired either through single plane acquisition or by acquiring a *Z*-stack. Epifluorescence *Z*-stacks were deconvolved to reduce out-of-focus light and improve contrast and resolution, while single slice images remained untouched. For confocal imaging, background correction was employed. Images were analyzed using ImageJ/FIJI (NIH, MD) or Volocity. For nuclear to cytosolic TFEB ratio, regions of interest were drawn around the nucleus and the cytosol, and the intensities within these regions were then measured. After background subtraction, the ratio of nuclear to cytosol fluorescence was calculated.

## Quantitative real-time PCR

RAW macrophages were infected with late-log or stationary phase *Salmonella* for 2, 5, and 20 h as described above. RNA was then extracted using the GeneJet RNA Purification Kit (Thermo Fisher Scientific) and transcribed to cDNA using the iScript Reverse Transcription Supermix for RT-qPCR (Bio-Rad). TaqMan gene expression assays were carried out using 50 ng cDNA, the TaqMan Gene Expression Master Mix (Applied Biosystems) and probes against murine GAPDH, LC3, Cathepsin-D, and Lamp1 (Table 1) . Data were normalized to the uninfected, wild-type RAW macrophages for each gene target.

**TABLE 1** TaqMan gene expression probes for each gene

| Gene name | Assay ID | Chromosome location |
|---|---|---|
| Microtubule-associated protein 1 light chain 3 beta (LC3) | Mm00782868_sH | Chr.8: 121590468–121598047 |
| Lysosomal-associated membrane protein 1 (Lamp1) | Mm01217070_m1 | Chr.8: 13159134–13175338 |
| Cathepsin D | Mm00515586_m1 | Chr.7: 142375916–142387870 |
| GAPDH | Mm99999915_g1 | Chr.6: 125161338–125166511 |

## Cell lysates, SDS-PAGE, and Western blotting analysis

After specific cell treatments, whole cell lysates were prepared using 2× Laemmli buffer containing phosSTOP and protease inhibitor cocktail (Roche, Mississauga, ON). Lysates were passed through a 27-gauge needle 10 times, then heated at 65°C for 15 min. For total TFEB levels, lysates were resolved by SDS-PAGE using a 10% acrylamide resolving gel. Proteins were transferred to a PVDF membrane, blocked with 5% BSA, incubated with primary rabbit anti-TFEB antibodies (Bethyl Laboratories, Montgomery, TX) or primary rabbit anti-β-actin antibodies (Cell Signaling Technology, Danvers, MA) at a dilution of 1:1,000, followed by HRP-conjugated goat anti-rabbit secondary antibodies (Cell Signaling Technology) at 1:5,000 dilution. Protein levels were detected via Clarity enhanced chemiluminescence reagent (Bio-Rad Laboratories, Mississauga, ON) visualized by a ChemiDoc Touch Imaging system (Bio-Rad). Blot intensity quantification was accomplished using ImageLab software (Bio-Rad) and its volumetrics and background subtraction functions. Total TFEB protein levels were normalized to®-actin.

## Phos-tag-PAGE and Western blotting analysis

For analysis of protein phosphorylation, lysates were prepared as described above. Lysates were then run on a 7.5% acrylamide resolving gel containing 50 µM SuperSep Phos-tag (Wako Chemicals, Richmond, VA). Proteins were separated at 15 mA and the gel was then washed as per the manufacturer's instructions. Proteins were transferred onto a PVDF membrane and visualized as described above. Phosphorylated proteins experience decreased mobility in Phos-tag gels relative to less phosphorylated isoforms. To analyze changes in phosphorylation of TFEB, we marked the migration front of TFEB in resting cells, then measured the volume of TFEB signal upstream and downstream of this front for each condition using ImageLab (Bio-Rad). We then measured the ratio of the downstream volume (less phosphorylated) to the upstream volume (more phosphorylated) to assess the degree of TFEB phosphorylation in each condition.

## Statistical analyses

All experiments were repeated independently at least three times with the exact number shown in the figure and/or legend. For bacterial survival, each condition had two or three technical replicates. For microscopy analysis, the number of individual cells scored is indicated in the corresponding figure legend. The mean of independent experiments is indicated along with either standard deviation or standard error of the mean, as appropriate. As indicated, sometimes data were normalized to a control condition to account for experimental variation. Unless otherwise stated, data were assumed to be normally distributed and/or have equal variability. Unless otherwise stated, data were subjected to analysis of variance (ANOVA) followed by a post hoc test recommended by GraphPad Prism such as Tukey's, Dunnett's, or Sidak's test. Data were matched by independent repeats and by concurrent conditions. A $P$ value of <0.05 was considered significant.

## ACKNOWLEDGMENTS

This research was made possible with financial support to R.J.B. from the Canada Research Chairs Program (950-232333), The Early Researcher Award from the Government of Ontario (ER13-09-042), The John Evans LOF Program from the Canada Foundation for Innovation (32957), the Research Instruments and Innovation Grant Program from Natural Sciences and Engineering Council of Canada (EQPEQ 458501-2014), and contributions from Toronto Metropolitan University (Formerly Ryerson University). The J.B.M. lab contributions were funded by the Discovery Program from the Natural Sciences and Engineering Council of Canada (RGPIN/04679-2015).

## AUTHOR AFFILIATIONS

[1]Department of Chemistry and Biology, Toronto Metropolitan University, Toronto, Ontario, Canada

[2]Molecular Science Graduate Program, Toronto Metropolitan University, Toronto, Ontario, Canada

## AUTHOR ORCIDs

Roberto J. Botelho http://orcid.org/0000-0002-7820-0999

## FUNDING

| Funder | Grant(s) | Author(s) |
| --- | --- | --- |
| Gouvernement du Canada \| Natural Sciences and Engineering Research Council of Canada (NSERC) | RGPIN/04679-2015 | Joseph B. McPhee |
| Gouvernement du Canada \| Natural Sciences and Engineering Research Council of Canada (NSERC) | EQPEQ 458501-2014 | Roberto J. Botelho |
| Canada Research Chairs (Chaires de recherche du Canada) | 950-232333 | Roberto J. Botelho |
| MDECEC \| Ontario Ministry of Research and Innovation (MRI) | ER13-09-042 | Roberto J. Botelho |
| Canada Foundation for Innovation (CFI) | 32957 | Roberto J. Botelho |
| Toronto Metropolitan University (TMU) | | Joseph B. McPhee |
| | | Roberto J. Botelho |

## AUTHOR CONTRIBUTIONS

Subothan Inpanathan, Conceptualization, Formal analysis, Investigation, Methodology, Visualization, Writing – original draft, Writing – review and editing | Erika Ospina-Escobar, Formal analysis, Methodology, Visualization, Writing – review and editing | Vanessa Cruz Li, Investigation, Visualization | Zainab Adamji, Investigation, Visualization, Writing – review and editing | Tracy Lackraj, Investigation, Visualization, Writing – review and editing | Youn Hee Cho, Formal analysis, Investigation, Methodology | Natasha Porco, Investigation, Methodology | Christopher H. Choy, Investigation, Visualization | Joseph B. McPhee, Funding acquisition, Resources, Supervision, Writing – review and editing | Roberto J. Botelho, Conceptualization, Formal analysis, Funding acquisition, Resources, Supervision, Writing – original draft, Writing – review and editing

## ADDITIONAL FILES

The following material is available online.

Open Peer Review

**PEER REVIEW HISTORY (review-history.pdf).** An accounting of the reviewer comments and feedback.

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
