## [Reviewer comments · Microbiology Spectrum]

Microbiology Spectrum

Salmonella actively modulates TFEB in murine macrophages in a growth-phase and time-dependent manner

Subothan Inpanathan, Erika Ospina-Escobar, Vanessa Li, Zainab Adamji, Tracy Lackraj, Youn Hee Cho, Natasha Porco, Christopher Choy, Joseph McPhee, and Roberto Botelho

Corresponding Author(s): Roberto Botelho, Toronto Metropolitan University

Review Timeline:

Submission Date:	December 4, 2022
Editorial Decision:	January 18, 2023
Revision Received:	September 29, 2023
Accepted:	November 1, 2023

Editor: Stacey Gilk

Reviewer(s): Disclosure of reviewer identity is with reference to reviewer comments included in decision letter(s). The following individuals involved in review of your submission have agreed to reveal their identity: Eun-Kyeong Jo (Reviewer #1); Ravi Manjithaya (Reviewer #2)

Transaction Report:

DOI: <https://doi.org/10.1128/spectrum.04981-22>

January 18, 2023

Dr. Roberto J. Botelho
Toronto Metropolitan University
Toronto
Canada

Re: Spectrum04981-22 (Salmonella actively modulates TFEB in murine macrophages in a growth-phase and time-dependent manner)

Dear Dr. Roberto J. Botelho:

Thank you for submitting your manuscript to Microbiology Spectrum. I have received the reviews; they agree that this study addresses an important question in field and provides insight into the role of TFEB during Salmonella infection. While Spectrum does not necessarily require mechanism, the reviewers raise valid questions regarding the downstream impact on Salmonella when TFEB activation is delayed, as well as how your findings fit into the broader context of infection. Reviewer #2 also suggested revisions to the graphs to improve readability.

Link Not Available

Sincerely,

Stacey Gilk

Journals Department
Reviewer comments:

Reviewer #1 (Comments for the Author):

The paper "Salmonella actively modulates TFEB in murine macrophages in a growth-phase and time-dependent manner" by Inpanathan et al. showed that the effect of Salmonella on TFEB was affected by the bacterial growth conditions and time of infection, whereby stationary, but not late-log bacteria, delayed TFEB activation. Some data are interesting; however, the paper suffers from the lack of detailed mechanisms. Here are the major comments:

1. If there are differences between stationary and late-log bacteria, which genes are differentially induced or suppressed between the two growth phases of bacteria?
2. Inactivation of the SPI-1 and SPI-2 T3SS of *S. Typhimurium* abrogated the inhibition of TFEB-nuclear translocation. The same effect was also seen when the master regulator, *phoP*, or the secreted effectors *sopD2* or *sifA* were deleted. What are the mechanisms by which these effectors modulate the TFEB nuclear translocation? Although the findings are interesting, the paper suffers from the lack of a mechanism.
3. Interestingly, IgG-primed phagocytosis, but not resting macrophages, restricted intracellular *Salmonella*, however, the mechanisms are vague.

Reviewer #2 (Comments for the Author):

Summary

The study by Inpanathan et al 'Salmonella actively modulates TFEB in murine macrophages in a growth-phase and time dependent manner' attempts to understand the effect of bacterial growth conditions on TFEB activation. The authors have clearly shown that infection with stationary phase *Salmonella* exhibits a delay in TFEB activation. The study also highlights the importance of *Salmonella* virulence factors in causing this delay. While it is an interesting finding that the metabolic status of *Salmonella* modulates TFEB activation, the study lacks in exploring the downstream consequence of this delayed TFEB activation. As TFEB is considered as the master regulator of lysosomes and autophagy, this study provides further understanding of the regulation of this important transcription factor.

Major comments

1. The major question that the authors have not addressed is 'What is the consequence of delayed TFEB activation'? The readouts for most of the key experiments are only microscopic analysis of TFEB and not taking into consideration the downstream effect of TFEB activation.
 - a) What happens to the expression of lysosomal and autophagy target genes after TFEB activation in stationary phase grown *E. coli* and *Salmonella*?
 - b) There is immediate TFEB translocation during infection with log phase *Salmonella*. Does this correspond to increased autophagy capture compared to stationary phase *Salmonella*?
2. It is surprising that authors do not see any difference in replication of *Salmonella* in TFEB $-/-$ macrophages. Is it cell type specific? As macrophages have multiple other mechanisms (antibacterial peptides/ROS) to prevent bacterial infection. It is therefore essential for authors to check the essentiality of TFEB in epithelial cells where lysosome-mediated killing and autophagy are important defence mechanisms against bacteria. If possible, intestinal epithelial cells that are relevant for *Salmonella* infection could be used.
3. The authors should discuss in detail the physiological relevance of this study during in vivo infection. Especially, how the growth conditions of bacteria play a role during in vivo *Salmonella* infection and replication.

Minor comments

1. Kindly revise the manuscript for typographical errors. For example, in figure 2, 'TFEB' is labelled as 'TFEG'.
2. It will be easy to understand if graph legends are consistent. In certain graphs such as figure 1B-D both colour and symbol (circle/square/triangle) are different whereas in other graphs like 2B-D only colour is different.
3. Maintain consistency in representing significance in graphs. Only figure 8 has 'p' values mentioned.

Staff Comments:

Preparing Revision Guidelines

For complete guidelines on revision requirements, please see the journal Submission and Review Process requirements at <https://journals.asm.org/journal/Spectrum/submission-review-process>. **Submissions of a paper that does not conform to**

Microbiology Spectrum guidelines will delay acceptance of your manuscript. "

Please return the manuscript within 60 days; if you cannot complete the modification within this time period, please contact me. If you do not wish to modify the manuscript and prefer to submit it to another journal, please notify me of your decision immediately so that the manuscript may be formally withdrawn from consideration by Microbiology Spectrum.

The paper “Salmonella actively modulates TFEB in murine macrophages in a growth-phase and time-dependent manner” by Inpanathan et al. showed that the effect of Salmonella on TFEB was affected by the bacterial growth conditions and time of infection, whereby stationary, but not late-log bacteria, delayed TFEB activation. Some data are interesting; however, the paper suffers from the lack of detailed mechanisms. Here are the major comments:

1. If there are differences between stationary and late-log bacteria, which genes are differentially induced or suppressed between the two growth phases of bacteria?
2. Inactivation of the SPI-1 and SPI-2 T3SS of *S. Typhimurium* abrogated the inhibition of TFEB-nuclear translocation. The same effect was also seen when the master regulator, *phoP*, or the secreted effectors *sopD2* or *sifA* were deleted. What are the mechanisms by which these effectors modulate the TFEB nuclear translocation? Although the findings are interesting, the paper suffers from the lack of a mechanism.
3. Interestingly, IgG-primed phagocytosis, but not resting macrophages, restricted intracellular Salmonella, however, the mechanisms are vague.

Point-by-point rebuttal

Editorial comments

"I have received the reviews; they agree that this study addresses an important question in field and provides insight into the role of TFEB during Salmonella infection. While Spectrum does not necessarily require mechanism, the reviewers raise valid questions regarding the downstream impact on Salmonella when TFEB activation is delayed, as well as how your findings fit into the broader context of infection. Reviewer #2 also suggested revisions to the graphs to improve readability."

Rebuttal: We would like to thank you and to thank our peers for the constructive feedback of the original work. We acknowledge that we do not define a mechanism by which *Salmonella* manipulates TFEB but agree that understanding the functional consequences of TFEB modulation by *Salmonella* is important. As enumerated below, we have assessed several measures of downstream impact by i) measuring the expression of genes previously connected to TFEB by qRT-PCR; ii) quantifying the effect on autophagy and xenophagy, and iii) extending studies of *Salmonella* survival and TFEB activation to a non-macrophage cell line. Details of these experiments and findings are summarized below.

Reviewer 1 comments

1. The paper "Salmonella actively modulates TFEB in murine macrophages in a growth-phase and time-dependent manner" by Inpanathan et al. showed that the effect of Salmonella on TFEB was affected by the bacterial growth conditions and time of infection, whereby stationary, but not late-log bacteria, delayed TFEB activation. Some data are interesting; however, the paper suffers from the lack of detailed mechanisms. Here are the major comments:

- 1. If there are differences between stationary and late-log bacteria, which genes are differentially induced or suppressed between the two growth phases of bacteria?*

To determine if stationary and late-log *Salmonella* infection differed in the expression of host genes, we measured the expression of LC3, LAMP1, and cathepsin D by qRT-PCR in infected cells over 2, 5 and 20 h post-infection. These are canonical lysosomal and autophagy genes that can be controlled by TFEB under at least some specific conditions. We also tested if any such changes caused by infection with *Salmonella* at different growth phases was dependent on TFEB/TFE3 by using wild-type, *tfeb*^{-/-} (SKO), and *tfeb*^{-/-} *tfe3*^{-/-} (DKO). For disclosure, we did all these conditions simultaneously at least three independent times, but because of the number of possible combinations, we re-analysed the same data in different ways to address specific questions and to make analysis manageable.

- i) First, we didn't observe changes in basal level of these genes in resting wild-type, SKO, and DKO macrophages (Fig. 8A).

- ii) Second, we then tracked expression of LAMP1, cathepsin D, and LC3 mRNAs in wild-type and mutant macrophages infected with late-log vs. stationary-grown *Salmonella* over 2, 5, and 20 h of infection.
- a. Within the parameters we tested, we didn't observe any differences in LAMP1 mRNA expression among any condition (Fig. 8B, C), suggesting that *Salmonella* infection and TFEB expression do not change LAMP1 mRNA levels within 20 h post-infection.
 - b. For cathepsin D, the pattern was more complex. We observed a relative increase in cathepsin D mRNA in wild-type macrophages infected for 20 h with stationary or late-log *Salmonella* (Fig 8D). Thus, *Salmonella* infection seems to modulate cathepsin D expression. Interestingly, SKO macrophages did not exhibit an increase in cathepsin D mRNA after infection with either *Salmonella* condition, but we cannot conclude that TFEB is required since basal levels of mRNA appeared to be higher overall (though not statistically significant relative to wild-type; Fig. 8D). Moreover, when we compare mRNA levels at 20 h infection between wild-type, SKO, and DKO macrophages, we did not observe a difference (Fig. 8E). Thus, *Salmonella* infection seems to promote cathepsin D expression, but this may not depend on growth phase or TFEB/TFE3.
 - c. We then assessed expression of LC3 mRNA. Here, there was no difference in mRNA expression in wild-type macrophages infected with *Salmonella* (Fig. 8F). However, late-log *Salmonella* infection of both SKO and DKO macrophages led to a statistically significant increase in LC3 mRNA levels after 20 h of infection. This was not observed in mutant macrophages infected with stationary-grown *Salmonella* (Fig. 8F, G). These data suggest that *Salmonella* growth phase can affect expression of LC3 mRNA in macrophages in a way that is promoted by the absence of TFEB and/or TFE3, not dependent on it – which was not as expected. This again suggests a complex interplay that likely involves other factors.
 - d. While we present these data, we also acknowledge that there are challenges in the assays we chose given the heterogeneity of infection conditions. We propose that single-cell assays that examine gene expression activity may be better suited to determine differences caused by *Salmonella* growth phase and TFEB/TFE3 instead of population-based methods like qRT-PCR. Given that we only examined three models, transcriptomics may also be suitable in the long-term. These statements can be found starting in line 450.

2. *Inactivation of the SPI-1 and SPI-2 T3SS of S. Typhimurium abrogated the inhibition of TFEB-nuclear translocation. The same effect was also seen when the master regulator, phoP, or the secreted effectors sopD2 or sifA were deleted. What are the mechanisms by which these effectors modulate the TFEB nuclear translocation? Although the findings are interesting, the paper suffers from the lack of a mechanism.*

Thank you for the feedback. We agree that revealing the mechanism by which these effectors modulate TFEB would be interesting. However, we are currently constrained in

our resources to further build on these observations. Thus, we believe that it is beneficial to disseminate these observational findings to encourage the community to resolve said mechanisms. Our main goal with this work was to disseminate that *Salmonella* manipulates TFEB in a context-dependent manner, which may guide future work by better defining key parameters. We do discuss possible mechanisms by how these effectors might modulate TFEB in the discussion, between lines 429-439.

3. *Interestingly, IgG-primed phagocytosis, but not resting macrophages, restricted intracellular Salmonella, however, the mechanisms are vague.*

We have previously demonstrated that phagocytosis of IgG-beads engages TFEB, increases lysosome genes in a TFEB-dependent manner, and boosted lysosomal and bactericidal activity (Gray *et al.*, 2016). In that work, we provided evidence that phagosome-lysosome fusion triggered Ca²⁺ released via the TPRML1 lysosomal channel to activate TFEB. We now explicitly discuss this starting in lines 334, 381, and 472.

Reviewer 2 comments

1. *The major question that the authors have not addressed is 'What is the consequence of delayed TFEB activation'? The readouts for most of the key experiments are only microscopic analysis of TFEB and not taking into consideration the downstream effect of TFEB activation.*

a) What happens to the expression of lysosomal and autophagy target genes after TFEB activation in stationary phase grown E. coli and Salmonella?

b) There is immediate TFEB translocation during infection with log phase Salmonella. Does this correspond to increased autophagy capture compared to stationary phase Salmonella?

A) Thank you for raising this issue, which was also asked by Reviewer 1. As such, please see our response above under Reviewer 1, point 1.

B) We expressed GFP-mCherry-LC3 reporter to simultaneously test autophagy levels, autophagosome maturation, and xenophagy. Wild-type RAW cells, *tfeb*^{-/-} (SKO) and *tfeb*^{-/-} *tfe3*^{-/-} (DKO) macrophages were infected with either stationary-grown or late-log grown *Salmonella* over 2, 4, and 6 h. Our data are presented in Figures 9 and 10.

- i. First, we didn't observe a difference in autophagosome number (total mCherry puncta) or autophagosome maturation (mCherry-positive, GFP-dim puncta) in resting wild-type, SKO, and DKO macrophages, suggesting that basal autophagy in unstressed cells is similar (Fig. 9A-D, Fig. 10A).
- ii. Second, focusing on total LC3-mcherry puncta (total autophagosomes), we observed that wild-type macrophages infected with late-log *Salmonella*

had reduced number of autophagosomes relative to macrophages infected with stationary-grown *Salmonella* (Fig. 9E). However, this difference remained true in SKO and DKO macrophages, suggesting that TFEB and/or TFE3 may not contribute to this disparity (Fig. 9F, G; though, judging from p values, the difference may actually be enhanced in the mutant macrophages). Overall, it seems that late-log *Salmonella* infection may repress autophagosome formation relative to stationary-grown *Salmonella*, but this was not readily dependent on TFEB/TFE3.

- iii. In terms of autophagosome maturation and xenophagy, we could not detect a statistically significant difference in macrophages infected with stationary vs. late-log grown *Salmonella* or between wild-type and mutant macrophages. These data are in Figure 10.
 - iv. Thus, admittedly, while our experiments suggest differences between late-log vs. stationary *Salmonella* in some situations, these are not readily explainable by different TFEB dynamics we observed earlier.
2. *It is surprising that authors do not see any difference in replication of Salmonella in TFEB -/- macrophages. Is it cell type specific? As macrophages have multiple other mechanisms (antibacterial peptides/ROS) to prevent bacterial infection. It is therefore essential for authors to check the essentiality of TFEB in epithelial cells where lysosome-mediated killing and autophagy are important defence mechanisms against bacteria. If possible, intestinal epithelial cells that are relevant for Salmonella infection could be used.*
- i. Unfortunately, we did not have access to an intestinal epithelial cell line that was deleted for TFEB. However, HeLa cells have been used as a model for infection of *Salmonella* and we had access to wild-type HeLa cells and triple knockout (*tfeb*^{-/-} *tfe3*^{-/-} *mitf*^{-/-}) HeLa cells. So, we used these cells to address the reviewers' question.
 - ii. We first examined wild-type HeLa cells and how infection with *Salmonella* at different growth phases affected endogenous TFEB translocation. Like macrophages, we observed that late-log *Salmonella* infection more readily promoted TFEB nuclear entry, though this required 4 h post-infection relative to control. Infection with stationary-grown *Salmonella* did not elicit a statistically significant change in TFEB nuclear signal. This is shown in Fig. 4.
 - iii. We then compared *Salmonella* survival in wild-type and TKO HeLa cells. During early infection (4 h post-infection), there were no differences in survival in *Salmonella* that depended on growth phase or TFEB (Fig. 11D). At 20 h post-infection, there was no difference in survival of stationary *Salmonella* infecting wild-type or TKO HeLa cells. However, we observed that late-log *Salmonella* survived more better in TKO HeLa cells than in wild-type, which suggests that TFEB related factors help repress *Salmonella* in a context dependent manner (Fig. 11D). It also suggests that macrophages may have additional mechanisms to repress *Salmonella* independently of TFEB.

3. *The authors should discuss in detail the physiological relevance of this study during in vivo infection. Especially, how the growth conditions of bacteria play a role during in vivo Salmonella infection and replication.*

We have commented on the physiological relevance of growth phase and Salmonella infections in vivo in our discussion highlighted (Lines 455-467). Overall, there is limited literature on the impact of *Salmonella* growth-phase on in vivo infection; some evidence suggests stationary-phase *Salmonella* delays animal mortality, is less susceptible to clearance early infection and has reduced colonization in peripheral organs. Some of these effects however, seem to be temporal in nature as the differences between growth-phases disappear with time, as we observe for repression of TFEB nuclear entry.

Minor comments

4. *Kindly revise the manuscript for typographical errors. For example, in figure 2, 'TFEB' is labelled as 'TFEG'.*

Fixed.

5. *It will be easy to understand if graph legends are consistent. In certain graphs such as figure 1B-D both colour and symbol (circle/square/triangle) are different whereas in other graphs like 2B-D only colour is different.*

Most graphs are now internally consistent.

3. *Maintain consistency in representing significance in graphs. Only figure 8 has 'p' values mentioned.*

We chose to display p values for survival data (current Figure 11 and 12) because the magnitude of the p value may be informative to the reader in cases where the threshold was close to $p < 0.05$, but not under.

Re: Spectrum04981-22R1 (Salmonella actively modulates TFEB in murine macrophages in a growth-phase and time-dependent manner)

Dear Dr. Roberto J. Botelho:

As you can see from the reviewer's comments, there are still concerns from reviewer 2 regarding the lack of mechanism and impact on downstream bacterial infection. While these are valid criticisms, after careful evaluation, I agree that these are follow-up studies and the current manuscript fulfills Spectrum's requirements of high quality work that will be useful for the community.

Your manuscript has been accepted, and I am forwarding it to the ASM production staff for publication. Your paper will first be checked to make sure all elements meet the technical requirements. ASM staff will contact you if anything needs to be revised before copyediting and production can begin. Otherwise, you will be notified when your proofs are ready to be viewed.

Sincerely,
Stacey Gilk
Editor
Microbiology Spectrum

Reviewer #1 (Comments for the Author):

My concerns have been cleared.

Reviewer #2 (Comments for the Author):

As I said before, the authors have a potential story here but are not able to explain their results in the context of Salmonella infection. I am particularly concerned that the TFEB data is inconclusive, and the authors do not provide a satisfactory/alternative explanation or hypothesis.

I am sharing below the information that may be useful to the authors:

Concerns:

1) RT-PCR results: Lamp1, LC3, Cathepsin D: high variation and no conclusive results.

- 2) Not much effect of TFEB and TFE3 KO on Lamp1, LC3, Cathepsin D expression level. Alternative compensatory mechanisms?
- 3) Although repression of TFEB translocation by stationary phase Salmonella infection clearly shown, downstream effect not seen. No change in Salmonella growth & survival and Lamp1, LC3, Cathepsin D gene expression.
- 4) Story not complete. Mechanism of how the effector proteins manipulate host pathway to repress TFEB translocation not seen.
- 5) TFEB translocation repression by stationary phase bacteria shown satisfactorily by imaging. Phospho/dephospho levels do not match these findings (no effect on the ratio).

Additionally, the major and minor points that remain unresolved are also pasted below:

Major comments:

- 1) Is the effect of Salmonella infection on TFEB translocation burden dependent? Binning the cells based on no. of intracellular bacteria will shed light on this.
- 2) Is the intracellular localization of Salmonella (vacuolar/non-vacuolar) affected by growth-phase of infection?
- 3) Does the number of bacteria that enter the cells differ across growth phases? That can be a possible reason for differential TFEB translocation.
- 4) Line185-188: 'The absence of nuclear TFEB in HeLa cells infected with stationary-grown Salmonella may reflect the less invasive nature of these bacteria, which could affect uptake by non-phagocytes. However, TFEB remained predominantly cytosolic even in HeLa cells with larger number of engulfed Salmonella.' This statement can be supported by binning the cells based on the number of intracellular bacteria.
- 5) Is the TFEB suppressive function of sopD2, sifA and phoP growth-phase dependent? What is the effect of late-log phase infection with sopD2, sifA and phoP Salmonella mutants on TFEB translocation as compared to wt?
- 6) The authors are suggested to investigate the status of TFEB translocation upon infection with wt and mutants: sopD2, sifA and phoP in late-log and stationary phase at later time points of infection. This is especially relevant because the effectors of SPI-2 secretion system like SifA are highly expressed and extensively manipulate the host at later stages of infection.

Minor comments:

- 1) Line54: Salmonella actively promotes TFEB and Line55: compounds that promote TFEB seem vague. Can be replaced with 'Salmonella actively promotes TFEB expression/activity' and 'compounds that promote TFEB expression/activity'.
- 2) Figure 2A: TFEB channel can be shown separately along with merge panel for clear visualization of nuclear translocation.
- 3) Label figure 2C as 'Effect of time on stationary phase bacteria'
- 4) Check for typos: Line410: 'It is also not unknown'